# Dynamic patterns of information flow in complex networks

Uzi Harush[1] & Baruch Barzel[1]

Although networks are extensively used to visualize information flow in biological, social and technological systems, translating topology into dynamic flow continues to challenge us, as similar networks exhibit fundamentally different flow patterns, driven by different interaction mechanisms. To uncover a network's actual flow patterns, here we use a perturbative formalism, analytically tracking the contribution of all nodes/paths to the flow of information, exposing the rules that link structure and dynamic information flow for a broad range of nonlinear systems. We find that the diversity of flow patterns can be mapped into a single universal function, characterizing the interplay between the system's topology and its dynamics, ultimately allowing us to identify the network's main arteries of information flow. Counter-intuitively, our formalism predicts a family of frequently encountered dynamics where the flow of information avoids the hubs, favoring the network's peripheral pathways, a striking disparity between structure and dynamics.

[1] Department of Mathematics, Bar-Ilan University, Ramat-Gan 52900, Israel. Correspondence and requests for materials should be addressed to B.B. (email: baruchbarzel@gmail.com)

The recent years have witnessed major advances in our ability to map the structure of many natural and man-made complex systems[1–5], from social networks[6] and infrastructure systems[7, 8] to sub-cellular interaction mapping has uncovered several universal characteristics, observed across networks of vastly different domains, such as the small-world phenomenon[11] or the commonly observed fat-tailed degree[12, 13] and weight[14, 15] distributions. Our ultimate goal, however, is to translate these structural characteristics into functional predictions pertaining to the system's dynamic behavior[16–18]. For instance, we wish to use the topology of the gene regulatory network to gain insight into the functional pathways along which genetic information is transmitted[19, 20], or to translate the social network topology into predictions on the propagation of influence through social ties[21, 22]. The problem is that information flow is not determined solely by the static network topology, but also by the nonlinear dynamics characterizing the interactions between the nodes[18, 23]. Hence, the same network may exhibit fundamentally different patterns of information flow under different dynamics: epidemic spread, ecological interactions, or genetic regulation.

To observe these patterns we employ here a perturbative approach, a fundamental tool to uncover information propagation[24], specifically applicable in the context of network dynamics[17, 18, 25, 26]. We then analytically track the propagation of signals between nodes, identifying the main pathways through which these signals penetrate the network. Our results show that despite the diversity of potential interaction mechanisms, the patterns of information flow are governed by universal laws that can be directly linked to the system's microscopic dynamics.

## Results

**Quantifying information flow.** We consider a system of $N$ components (nodes) linked via a weighted and directed network $A_{ij}$. Each node is characterized by a time dependent activity $x_i(t)$, $i = 1, \ldots, N$, whose meaning depends on the specific application: for instance, the concentration of a protein in a cellular network, the abundance of a species in an ecological networks or the probability of infection of an individual in a social network. The system's dynamics is driven by[18, 27]

$$\frac{dx_i}{dt} = M_0(x_i) + \sum_{j=1}^{N} A_{ij} M_1(x_i) M_2(x_j), \qquad (1)$$

where the first term on the r.h.s. accounts for $i$'s self-dynamics, and the second term captures the impact of $i$'s interacting partners. By appropriately selecting the nonlinear functions

$\mathbf{M} = (M_0(x), M_1(x), M_2 1)$ provides a rather general description of complex system dynamics, including frequently used models to describe the behavior of social[21, 28–30], biological[25, 31–33] and technological[34, 35] systems (Table 1). Note that in (1) the weighted link $A_{ij}$ represents the rate of incoming influence from $x_j$ to $x_i$, hence $A_{ij} = A_{i \leftarrow j}$, a directed link outgoing from $j$, incoming to $i$.

We can track the propagation of a signal through the system (1) by following how a local perturbation in the steady-state activity of a source node $n$ impacts the activities of all remaining nodes in the system, giving rise to the linear response matrix[17, 18]

$$G_{mn} = \left| \frac{dx_m/x_m}{dx_n/x_n} \right| = \left| \frac{d\log x_m}{d\log x_n} \right|. \qquad (2)$$

The terms of $G_{mn}$ capture the level of information spread form the source $n$ to a specific target node $m$. Summing over all targets, we obtain the total capacity of information distributed from $n$ throughout the network as

$$Z_n = \sum_{m=1}^{N} G_{mn}, \qquad (3)$$

capturing the cumulative response of the system to the signal $dx_n$.

Consider the contribution of an intermediate node $i$ to $Z_n$: first the signal $dx_n$ reaches $i$, then $i$ responds by shifting its own activity by $dx_i$, in effect creating a new signal that helps propagate $dx_n$ to the rest of the network. If we now artificially set $dx_i = 0$, we freeze $i$'s activity, forcing it to remain unperturbed, and hence preventing it from propagating the signal $x_n$ onward. The result is $Z_n^{\{i\}}$, capturing the level of information spread from $n$ under the freezing of $x_i$, effectively blocking all flow of information $n \to m$ via pathways that pass though $i$ (Fig. 1a). More generally, we can freeze the flow through an entire network path, $\Pi = \{i, A_{ij}, j, A_{jk}, k, \ldots\}$, in which case we block the flow of information through a sequence of nodes and links, providing $Z_n^{\Pi}$. This allows us to quantitatively evaluate the contribution of $\Pi$ to the flow from the source $n$ as

$$\mathcal{F}_n^{\Pi} = \frac{Z_n - Z_n^{\Pi}}{Z_n}, \qquad (4)$$

capturing the fraction of $Z_n$ that was mediated through the $\Pi$ pathway. Averaging over all $n$ we obtain $\Pi$'s overall flow

$$\mathcal{F}_{\Pi} = \frac{1}{N} \sum_{n=1}^{N} \mathcal{F}_n^{\Pi}, \qquad (5)$$

**Table 1 Network dynamics**

| Dynamics | Equation | Symbol | $\omega$ | $\xi$ | Class |
|---|---|---|---|---|---|
| Population | $\frac{dx_i(t)}{dt} = -x_i^3(t) + \sum_{j=1}^{N} A_{ij} x_i^2(t)$ | $\mathbb{P}$ | $\frac{5}{3}$ | $\frac{2}{3}$ | Degree driven |
| Regulatory | $\frac{dx_i(t)}{dt} = -x_i(t) + \sum_{j=1}^{N} A_{ij} \frac{x_j^{\frac{1}{3}}(t)}{1+x_j^{\frac{1}{3}}(t)}$ | $\mathbb{R}_1$ | $\frac{2}{3}$ | $-\frac{1}{3}$ | Degree driven |
| Epidemic | $\frac{dx_i(t)}{dt} = -x_i(t) + \sum_{j=1}^{N} A_{ij}(1 - x_i(t)) x_j(t)$ | $\mathbb{E}$ | $0$ | $-1$ | Homogeneous |
| Biochemical | $\frac{dx_i(t)}{dt} = 1 - x_i(t) - \sum_{j=1}^{N} A_{ij} x_i(t) x_j(t)$ | $\mathbb{B}$ | $0$ | $-1$ | Homogeneous |
| Mutualistic | $\frac{dx_i(t)}{dt} = x_i(t)(1 - x_i(t)) + \sum_{j=1}^{N} A_{ij} x_i(t) \frac{x_j^2(t)}{1+x_j^2(t)}$ | $\mathbb{M}$ | $-1$ | $-2$ | Degree avert |
| Regulatory | $\frac{dx_i(t)}{dt} = -x_i(t) + \sum_{j=1}^{N} A_{ij} \frac{x_j^2(t)}{1+x_j^2(t)}$ | $\mathbb{R}_2$ | $-1$ | $-2$ | Degree avert |

We applied our formalism to six different types of dynamics of the form (1): $\mathbb{P}$ captures population dynamics through birth-death processes;[32] $\mathbb{R}_1$ captures regulatory dynamics, e.g., gene regulation, via the Michaelis–Menten model, with a Hill coefficient of $h = 1/3$[33]; $\mathbb{E}$ is the susceptible-infected-susceptible (SIS) model for epidemic spread;[28–30] $\mathbb{B}$ captures biochemical dynamics, e.g., protein-protein interactions, modeled using mass-action kinetics;[31, 41] $\mathbb{M}$ represents mutualistic interactions between species in an ecological network[42] and $\mathbb{R}_2$ is the same as $\mathbb{R}_1$ with a different Hill coefficient, $h = 2$. For each dynamics we also show $\omega$ (8) and $\xi$ (9), and its classification as degree-driven flow (red), homogeneous flow (green) or degree-averting flow (blue). A detailed description and analysis of all models appears in Supplementary Note 2

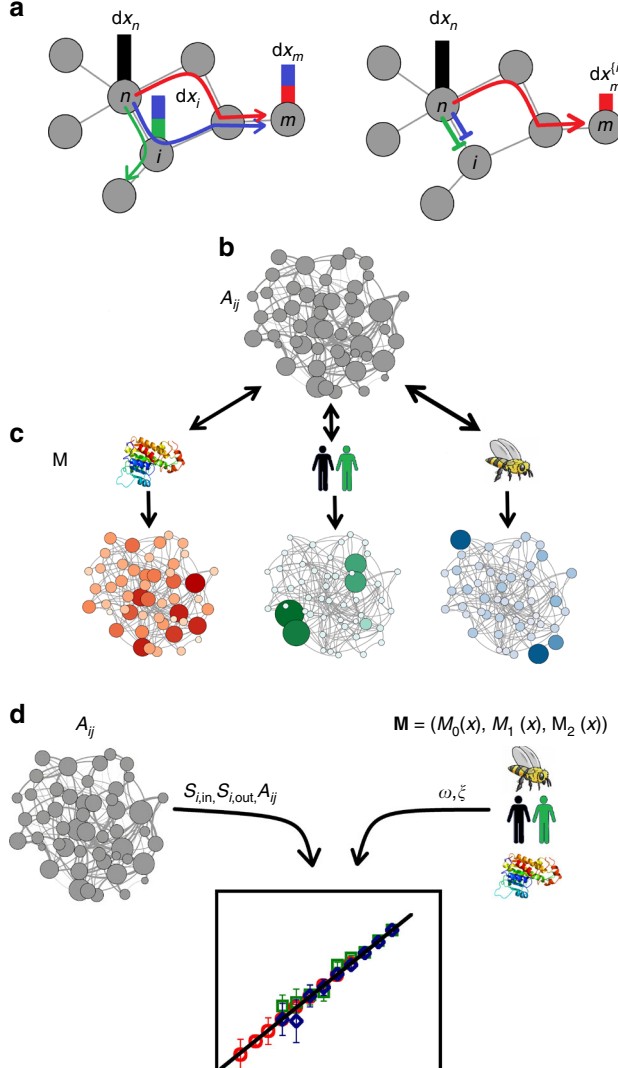

**Fig. 1** Observing and predicting the flow of information in a network environment. **a** The spread of information through the network is captured by the propagation of activity changes (signals) between network components (nodes). Here the activity of the source node $n$ is perturbed, giving rise to the signal $dx_n$ (black). The information from this signal reaches the target node $m$, whose response $dx_m$ is mediated through all relevant pathways (red, blue). By freezing the activity of the intermediate node $i$ (right), we block the blue pathway, resulting in a reduction of information flow in the network, and a diminished response $dx_m^{\{i\}}$, which now includes only the red portion, lacking the blue contribution. The diminished $dx_m^{\{i\}}$ allows us to quantify $i$'s contribution to the flow of information from $n$ to $m$, or, by summing over all pairs $m, n$ its overall contribution to the flow, $\mathcal{F}_i$. **b** The topology of a complex system is given by the weighted and directed network $A_{ij}$, a random network with arbitrary degree/weight distribution. **c** Similar networks may exhibit diverse flow patterns, depending on the microscopic interaction mechanisms between the nodes, as captured by $M = (M_0(x), M_1(x), M_2(x))$ in (1). Hence, different dynamics, e.g., biochemical, social or ecological, may lead to fundamentally different flow patterns (red, green, blue). **d** Our goal is to predict how topological characteristics ($A_{ij}$) interplay with the system's interaction dynamics (**M**) to produce the observed flow patterns. The topology is characterized by $S_{i,in}$, $S_{i,out}$ and the weights $A_{ij}$; the dynamics is captured by the exponents $\omega$ and $\xi$. Together they provide the flow through Eqs. (6) and (9). The diversity of flow patterns, i.e., red, green, and blue in **c**, can be captured by a single universal flow function

quantifying, systematically, the contribution of each pathway to the spread of signals ($G_{mn}$) throughout the network. In case $\mathcal{F}_\Pi \ll 1$, $\Pi$'s contribution to the flow of information in the system (1) is marginal; if, however, $\mathcal{F}_\Pi \to 1$, then almost all information flows through $\Pi$.

To place our proposed measure of flow (5) in context we emphasize the distinction between influence and flow. Most often, network components—nodes, links, pathways—are ranked according to their dynamic impact on the network, e.g., seeking the most influential nodes[36]. In the context of our current formalism, such impact is captured by the magnitude of $Z_n$ (3), namely the response of the system to $n$'s perturbation. However, most of the time a network component is not the source of information, but rather the mediator of the information that constantly travels between arbitrary locations on the network. For example, when a single gene $n$ out of $N \sim 10^4$ is perturbed, that gene is the only source of information, whereas the role of all remaining genes is to propagate $n$'s signal, supporting flow as mediators, not as sources. Hence $\mathcal{F}_\Pi$, designed to capture the efficiency of a pathway as a "pipe" rather that a source of information flow, provides a crucial, overlooked, metric of the ongoing dynamic role continuously played by all network components.

**Observing the patterns of flow**. To observe the diverse patterns of flow exhibited by (1) we constructed a set of model and empirical networks, capturing systems from a broad range of scientific domains, including weighted scale-free networks with scale-free weights (SF1—undirected, SF2—directed); protein interactions from human and yeast cells (Human PPI[37] and Yeast PPI[9]); two online social networks (UCIonline[38] and Epoch[39]) and a bipartite ecological network, capturing plant-pollinator relationships (ECO1, ECO2[40]). We then implemented six different types of frequently used dynamic models **M**, capturing diverse forms of interaction mechanisms: the susceptible-infected-susceptible model[21, 28–30] for epidemic spreading ($\mathbb{E}$), biochemical interactions via mass-action-kinetics[31, 41] ($\mathbb{B}$), mutualistic dynamics in ecology[42] ($\mathbb{M}$), population dynamics[32, 35] ($\mathbb{P}$), and genetic regulation as captured by the Michelis–Menten model[33, 43] ($\mathbb{R}_1$ and $\mathbb{R}_2$), all summarized in Table 1.

For each system we measured the flow through all nodes and edges, $\mathcal{F}_i$ and $\mathcal{F}_{ij}$, respectively. For $\mathcal{F}_i$ we selected $\Pi = \{i\}$ in (5), a path including a single node, and for $\mathcal{F}_{ij}$ we repeated the calculation with $\Pi = \{A_{ij}\}$, freezing sequentially all edges. Hence, we obtain the contribution of all individual nodes ($\mathcal{F}_i$) and edges ($\mathcal{F}_{ij}$) to the flow of information in the system. We find that the patterns of flow exhibit an extremely high level of diversity across the different systems, as expressed by the distinct size distribution of nodes (or width of edges) across the twenty-four layouts presented in Fig. 2. For instance, in Fig. 2a–f we show the flow patterns obtained by applying different dynamics (**M**) to the same network (SF1). It shows that despite the fact that $A_{ij}$ remains the same, the dynamic patterns of flow are highly distinctive. For $\mathbb{P}$ and $\mathbb{R}_1$ information flow is dominated by a few selected central nodes. In contrast, under $\mathbb{E}$ and $\mathbb{B}$ the same network exhibits a distributed flow, with almost all nodes equally contributing to the spread of information. Finally, $\mathbb{M}$ and $\mathbb{R}_2$ show yet another pattern of information flow, with a seemingly random scatter of flow hubs spread throughout the network. Such diversity is also observed for SF2 (Fig. 2g–l), or for the empirical networks, where the same topology exhibits profoundly different flow patterns, depending on the system's dynamics (Fig. 2m–x). Hence, the patterns of flow are a consequence not just of the topology, but of the intricate interplay between this topology and the system's interaction dynamics (Fig. 1b, c). Taken together, the twenty-four

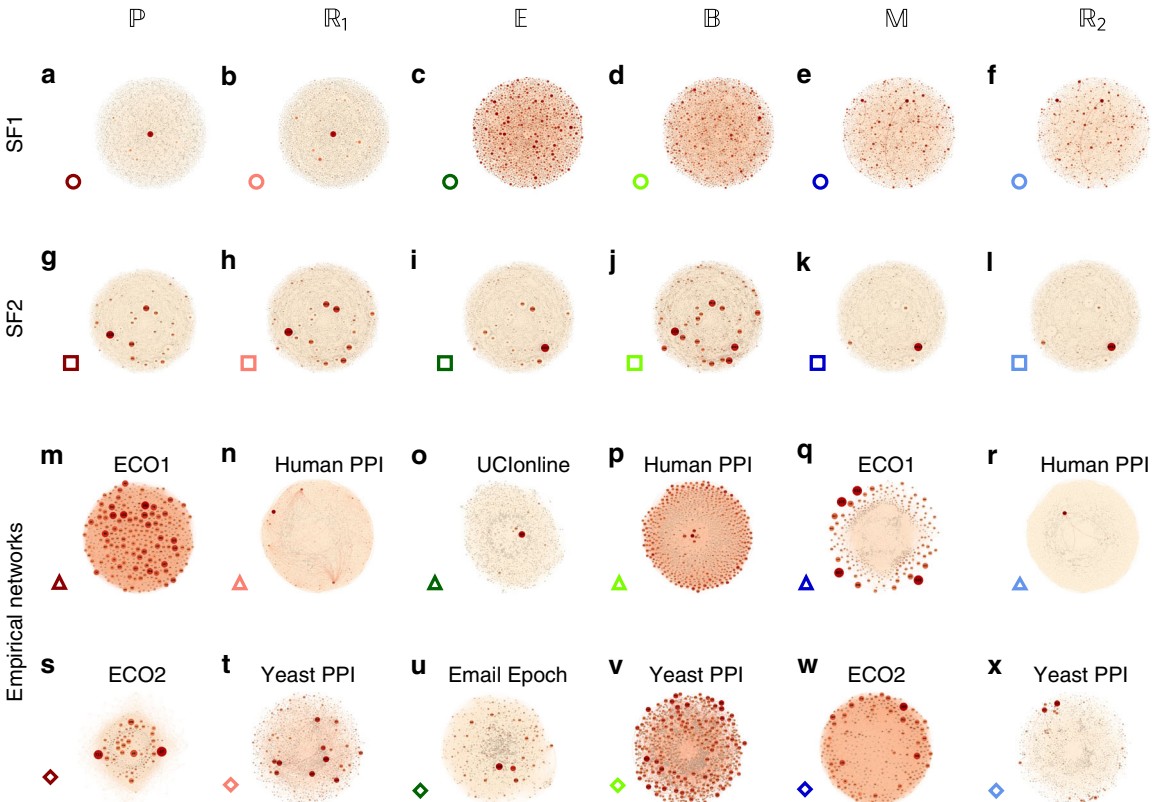

**Fig. 2** Dynamic information flow in model and real networks. We measured the flow (5) through all nodes in a set of model and real networks, on which we implemented the six dynamic models of Table 1. The size of each node and the width of each link are proportional to their flow $\mathcal{F}_i$ and $\mathcal{F}_{ij}$. **a–f** For SF1, a weighted network with scale-free degree/weight distributions, we observe highly diverse flow patterns across the different dynamic models: in $\mathbb{P}$ **a** and $\mathbb{R}_1$ **b** flow is dominated by extremely few nodes; in $\mathbb{E}$ **c** and $\mathbb{B}$ **d** flow is distributed over all nodes; in $\mathbb{M}$ **e** and $\mathbb{R}_2$ **f** we find again that only a selection of nodes dominates the flow, though different than the one dominating $\mathbb{P}$ and $\mathbb{R}_1$. Hence, different dynamics give rise to distinctive flow patterns despite using the same network. **g–l** Flow through SF2, a directed and weighted scale-free network. **m–x** For each dynamics we also observe the flow in relevant empirical networks: ecological dynamics ($\mathbb{P}, \mathbb{M}$) on plant-pollinator networks (ECO1, ECO2), sub-cellular dynamics ($\mathbb{B}, \mathbb{R}_1, \mathbb{R}_2$) on protein interaction networks (Human PPI, Yeast PPI), epidemic flow ($\mathbb{E}$) on social networks (UCIonline, Email Epoch). Together, 24 combinations of networks and dynamics, exhibit a seemingly unpredictable "zoo" of diverse flow patterns

networks of Fig. 2, demonstrate a highly diverse set of flow patterns, illustrating the extreme challenge in predicting the dynamics of information flow in complex systems.

**Predicting the system's flow patterns**. To understand the origins of the observed flow patterns we derive $\mathcal{F}_i$'s dependence on the network's degree distribution, by linking it with the in and out weighed degrees of all nodes, $S_{i,\text{in}} = \sum_{j=1}^{N} A_{ij}$ and $S_{i,\text{out}} = \sum_{j=1}^{N} A_{ij}^{\top}$. We show in Supplementary Note 1 that, on average, information flow scales with a node's in/out-degree as

$$\mathcal{F}_i \sim S_{i,\text{out}} S_{i,\text{in}}^{\omega-1}, \tag{6}$$

where the scaling exponent $\omega$ is fully determined by the system's dynamics $\mathbf{M}$. To understand the contribution of $\mathbf{M} = (M_0(x), M_1(x), M_2(x))$ we link $\omega$ in Supplementary Note 1 to the Hahn series expansion

$$M_2\left(W^{-1}(x)\right) = \sum_{\Gamma(n)} C_n x^{\Gamma(n)}, \tag{7}$$

where $W(x) = -M_1(x)/M_0(x)$ and $W^{-1}(x)$ denotes its inverse function. The Hahn[44] expansion (7) is a generalization of the Taylor expansion to allow for both negative and real powers; the powers $\Gamma(n)$ represent a well-ordered set in ascending order with $n$, namely $\Gamma(0)$ represents the leading power in the expansion of $M_2(W^{-1}(x))$, $\Gamma(1) > \Gamma(0)$ is the next power and so on. The

exponent $\omega$ in (6) can be linked directly to the system's dynamics via (7) as

$$\omega = \begin{cases} 1 - \Gamma(0) & \Gamma(0) \neq 0 \\ 1 - \Gamma(1) & \Gamma(0) = 0 \end{cases}, \tag{8}$$

hence $\omega$ is determined by the leading non-vanishing exponent in (7). While the specific value of $\omega$ depends on the dynamic model $\mathbf{M}$ ($\mathbb{P}, \mathbb{R}_1$, etc.) the formula (8) to extract it from a given model is universal, providing a step-by-step method for constructing the flow in (6). An explicit example is shown in Methods.

Equations (6–8), represent our first analytical result, exposing the rules that govern the flow of information in a complex network. The scaling exponent $\omega$ helps us link between the system's structure ($S_{i,\text{in}}, S_{i,\text{out}}$) and its dynamic patterns of information flow ($\mathcal{F}_i$), providing the connection we seek between the system's topology and its actual observed flow patterns (Fig. 1d). In other words, Eq. (6) helps us translate topological characteristics, such as the weighed in/out degrees, into dynamic insights pertaining to the flow of information, thus addressing a fundamental challenge of network science[17, 45].

Most importantly, our formalism predicts that the diversity observed in Fig. 2 is, in fact, rooted in a deep universality, expressed by the mapping of structure to dynamics that appears in (6). To test this, we revisit the "zoo" of twenty-four diverse flow patterns presented in Fig. 2 and confront the observed flow

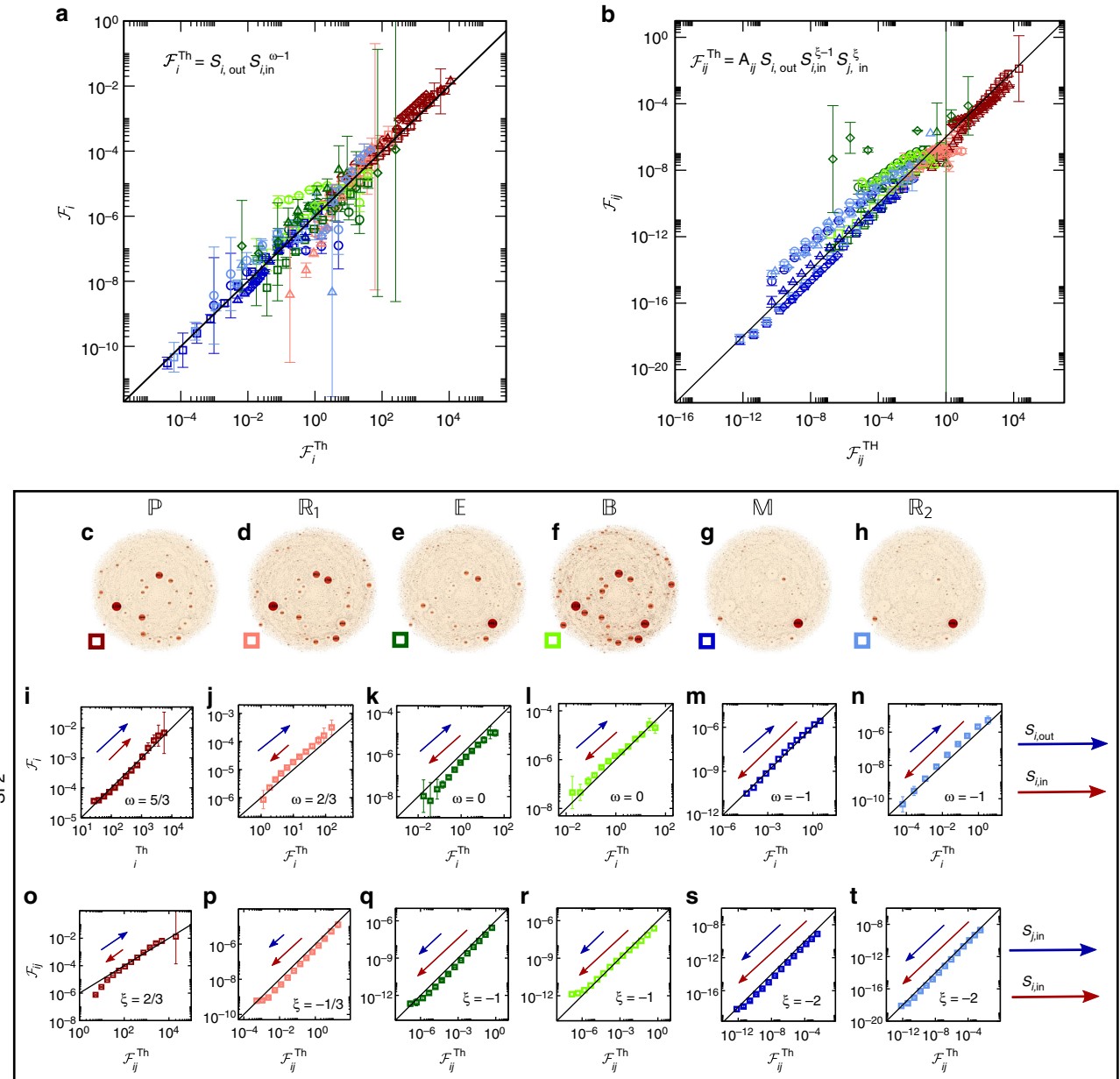

**Fig. 3** Predicting the observed flow patterns. We used Eqs. (6) and (9) to predict the flow of information in the twenty-four systems of Fig. 2, finding that despite their diverse behaviors, they all emerge from these two universal equations. **a** $\mathcal{F}_i$ vs. $\mathcal{F}_i^{Th} = S_{i,out} S_{i,in}^{\omega-1}$ for all layouts of Fig. 2. As predicted, all flow patterns condense around the single analytically derived function (linear solid line) of (6), with their diversity rooted in the different values of the dynamic exponent $\omega$ (8). The color/shape representing each network appears adjacent to the appropriate layout in Fig. 2, *e.g.*, SF1 with dynamics $\mathbb{P}$ (Fig. 2a) is represented by dark red circles, SF2 with $\mathbb{P}$ (Fig. 2g)—dark red squares, etc. **b** $\mathcal{F}_{ij}$ vs. $\mathcal{F}_{ij}^{Th}$, as predicted in (9), for all networks/dynamics of Fig. 2, showing an agreement that spans over sixteen orders of magnitude. **c–h** Revisiting the flow patterns of the directed scale-free network SF2 (identical to Fig. 2g–l). **i–n** $\mathcal{F}_i$ vs. $\mathcal{F}_i^{Th}$ as extracted from the SF2 layouts shown in **c–h**. The arrows point in the direction of large $S_{i,in}$ (red) and $S_{i,out}$ (blue). For instance, under $\mathbb{P}$ dynamics ($\omega = 5/3$) the flow increases with both in/out degrees, hence both arrows point upwards, contributing less to the flow. In contrast for $\mathbb{R}_1$ ($\omega = 2/3$), nodes with large $S_{i,in}$ are concentrated at the bottom left, contributing less to the flow. The strength of the effect is captured by the length of the arrow. Equation (6) rearranges the nodes, locating the in-hubs at the high flow limit (top right) or at the low flow limit (bottom left) depending on the value of $\omega$, providing qualitative insight on the characteristics that increase/decrease flow for each type of dynamics. **o–t** $\mathcal{F}_{ij}$ vs. $\mathcal{F}_{ij}^{Th}$ as obtained from SF2. The arrows point in the direction of large $S_{j,in}$ (blue) and $S_{i,in}$ (red). Details on all networks and dynamic simulations are outlined in Supplementary Note 3. Error bars represent 95% confidence intervals (Supplementary Note 3)

through all nodes $\mathcal{F}_i$ with our universal prediction $\mathcal{F}_i^{Th} = S_{i,out} S_{i,in}^{\omega-1}$, taking for each system the relevant $A_{ij}$ and the appropriate value of $\omega$, as predicted by (8). Strikingly, we find in Fig. 3a that despite their diverse and unpredictable behavior, all layouts of Fig. 2 collapse onto the universal linear plot (solid line) predicted by (6). This collapse demonstrates the predictive power

of our formalism, taking a set of fundamentally different systems, from gene regulation to online social networks, cast on extremely diverse networks, and showing that they are all driven by similar rules of information flow, encapsulated within the universal relationship (6).

To gain a better grip on the mapping of (6) we focus specifically of the flow patterns of SF2 (original layouts appearing in Fig. 2g–l and presented again for convenience in Fig. 3c–h). In Fig. 3i–n we present the collapse plot of $\mathcal{F}_i$ vs. $\mathcal{F}_i^{\text{Th}}$, this time only for SF2, showing each dynamics separately. Once again we observe the derived universality, in which all data collapses along the theoretically predicted solid lines. However, the important point here is that now we can observe how the role of all nodes changes across the different dynamics, as expressed through their

location in each of the six plots. For instance, in $\mathbb{P}$, where $\omega = 5/3$, Eq. (6) predicts that nodes with high $S_{i,\text{in}}$ contribute more to the flow, hence occupying the top right quadrant of Fig. 3i, as noted by the direction of the red arrow. In contrast, for $\mathbb{R}_1$ ($\omega = 2/3$, Fig. 3j) the flow negatively scales with $S_{i,\text{in}}$, concentrating the high in-degree nodes toward the bottom left quadrant, thus capturing the qualitative difference in flow patterns across the different models, as predicted by our theory. The effect becomes more dramatic as $\omega$ is decreased in $\mathbb{E}$, $\mathbb{B}$, $\mathbb{M}$ and $\mathbb{R}_2$, pushing the in-

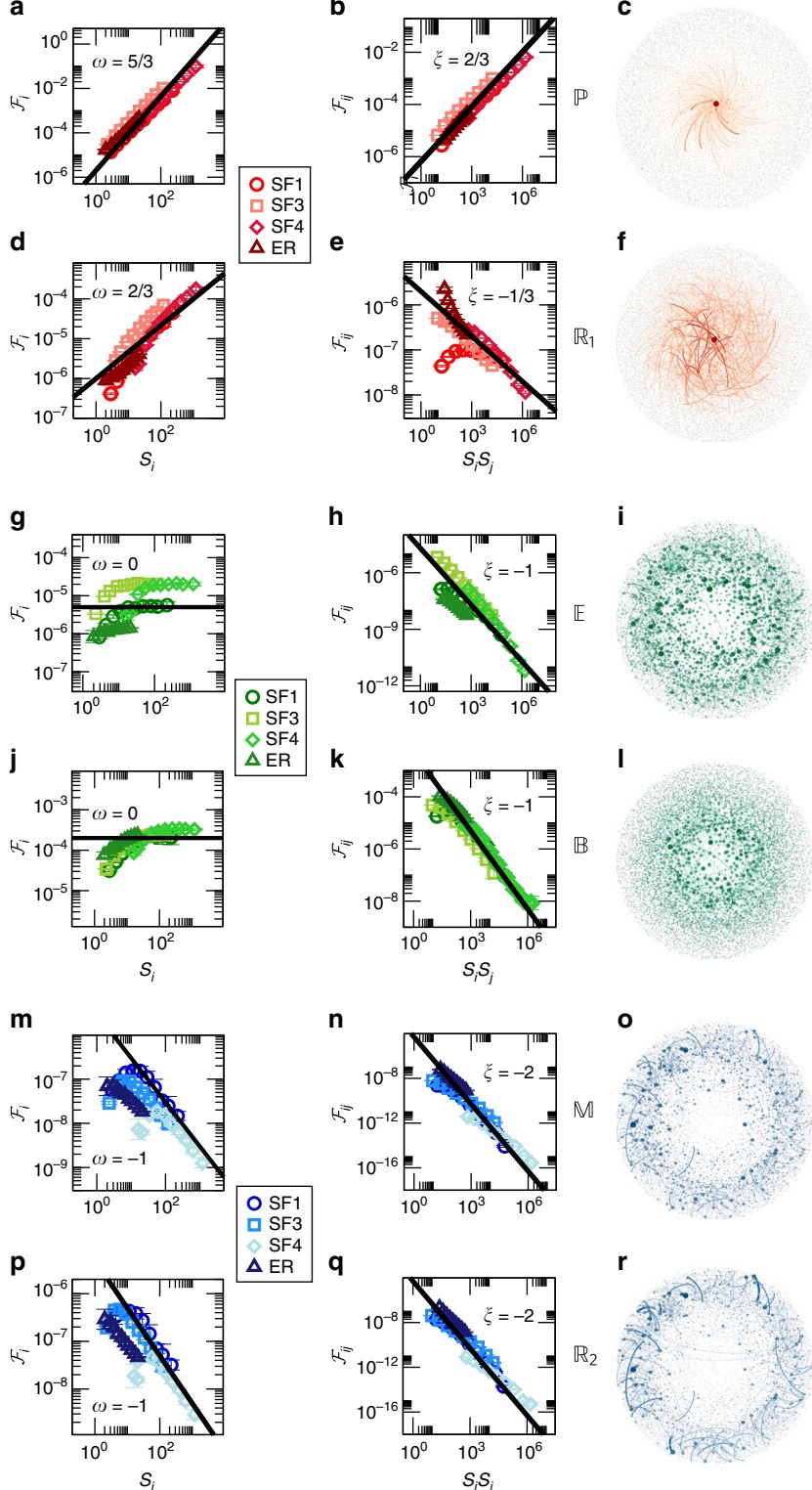

hubs further towards the limit of small $\mathcal{F}_i$ (bottom left), as symbolized by the length of the red arrows (Fig. 3k–n).

Next, we seek a similar universality to the one observed for $\mathcal{F}_i$ that can capture edge flow $\mathcal{F}_{ij}$. To observe this we show that, on average, the contribution of the $A_{ij}$ link to the propagation of information follows (Supplementary Note 1)

$$\mathcal{F}_{ij} \sim A_{ij}S_{i,\text{out}}^{\xi-1}S_{i,\text{in}}^{\xi}S_{j,\text{in}}^{\xi}, \qquad (9)$$

where $\xi = \omega - 1$ and $\omega$ is taken from (8). Hence $\mathcal{F}_{ij}$, associated with the link from $j$ to $i$ ($A_{i\leftarrow j}$) depends linearly on the link weight and on its target's outgoing weighted degree $S_{i,\text{out}}$, a rather expected interpretation of topology into information flow. The role of $i$ and $j$'s in-degrees, however, is more complex, affected also by the system's dynamics through $\xi$. To test this prediction we measured the $i,j$-flow, $\mathcal{F}_{ij}$, through all links in the networks of Fig. 2. Once again, in Fig. 3b, we observe that the seemingly random behavior observed in Fig. 2 hides a deep universality, in which all systems, despite their diverse topology/dynamics, condense around the predicted linear plot $\mathcal{F}_{ij} \sim \mathcal{F}_{ij}^{\text{Th}}$, where $\mathcal{F}_{ij}^{\text{Th}}$ is taken from our analytically predicted (9). The specific results obtained from SF2 are expanded in Fig. 3o–t, also indicating the roles of large $S_{i,\text{in}}$ (red arrows) and large $S_{j,\text{in}}$ (blue arrows) in each system.

The results of Fig. 3a, b and their expansion in Fig. 3i–t, expose an extremely robust universality, sustained across multiple orders of magnitude and diverse networks and dynamics, together— exposing an intricate balance: on the one hand, different systems exhibit highly distinct flow patterns, e.g., the different roles of in/out hubs in Fig. 3i–t. Yet, at the same time, all this richness, enabled by the topology/dynamics interplay, indeed the "zoo" of flow patterns observed in Fig. 2, is shown to originate from two universal analytically predictable sources, Eqs. (6) and (9).

**Universality classes of flow**. To obtain a deeper understanding of the implications of the derived universality we now focus on undirected networks, namely networks where all links are bi-directional ($A_{ij} \neq 0 \Leftrightarrow A_{ji} \neq 0$), but not necessarily weight-symmetric, hence potentially $A_{ij} \neq A_{ji}$. For randomly distributed weights, such networks have, on average, $S_{i,\text{in}} \sim S_{i,\text{out}} \equiv S_i$, which in (6) and (9) provide (Supplementary Note 1)

$$\mathcal{F}_i \sim S_i^{\omega} \qquad (10)$$

$$\mathcal{F}_{ij} \sim A_{ij}\left(S_iS_j\right)^{\xi}. \qquad (11)$$

These scaling relationships predict three highly distinctive dynamic universality classes:

Degree driven flow ($\omega > 0$, Fig. 4a–f, red). In case $\omega > 0$ the flow $\mathcal{F}_i$ in (10) increases with the weighted degree $S_i$, indicating that the flow of information is dominated by the high degree nodes. The greater is $\omega$, the more pronounced is the effect and

hence the more dominant is the role of the hubs. Equation (8) predicts that $\mathbb{P}$ and $\mathbb{R}_1$ belong to this class with $\omega = 5/3$ and $\omega = 2/3$, respectively. This analytical prediction is perfectly confirmed in Fig. 4a, d on the weighted scale-free network SF1 (circles).

Homogeneous flow ($\omega = 0$, Fig. 4g–l, green). In case $\omega = 0$ we have $\mathcal{F}_i$ independent of $S_i$, hence the contribution of the hubs to the flow of information is, on average, identical to that of the peripheral nodes. This represents homogeneous flow, where all nodes have almost similar contribution to the flow of information, independent of the network's often heterogeneous degree distribution. Using Eq. (8) we predict that $\mathbb{E}$ and $\mathbb{B}$ belong to this class. Indeed, Fig. 4g, j indicates that despite the three orders of magnitude diversity in the weighted degrees $S_i$, their contribution to the flow is largely homogeneous.

Degree-averting flow ($\omega < 0$, Fig. 4m–r, blue). For $\mathbb{M}$ and $\mathbb{R}_2$, Eq. (8) predicts $\omega = -1 < 0$, indicating that $\mathcal{F}_i$ decreases with $S_i$. Hence, strikingly, for this class of dynamics information flow tends to avoid the hubs, being dominated mainly by the majority of low degree nodes. Such counter-intuitive flow patterns, which favor the peripheral nodes, represent a highly unexpected outcome of prediction (8), and yet they are fully supported by the results presented in Fig. 4m, p, where $\mathcal{F}_i$ is in fact inversely proportional to $S_i$. These results, which defy the natural interpretation of topology to dynamics, highlight the importance of our formalism as well as its predictive strength, allowing us to expose such unique patterns of information flow.

Our formalism further predicts that $\omega$ and $\xi$, and consequently the three universality classes, are fully determined by the dynamics $\mathbf{M}$ through (8), independent of the network topology $A_{ij}$. Hence we implemented all six dynamic models (Table 1) on the relevant networks from Fig. 2. We also included several additional canonical model networks, such as an Erdős-Rényi network, and scale-free networks with binary (SF3) and normally distributed (SF4) weights, (in addition to SF1 that features scale-free distributed weights). We find that despite the diversity of the examined networks, the behavior of $\mathcal{F}_i$ and $\mathcal{F}_{ij}$ consistently exhibits the universal scaling predicted by (10) and (11), across all examined networks (Fig. 4).

Centralized vs. peripheral information flow. The analysis above helps us uncover the main arteries of information flow in a complex network, quantifying the contribution of each node/link, and hence of all pathways to the flow of information, as emerges from the interplay between the system's topology ($A_{ij}$, $S_i$) and its interaction dynamics ($\mathbf{M}$, $\omega$, $\xi$). To visualize this we used the scale-free SF1, presented in Fig. 2a–f, this time using a hub-central layout, in which the hubs (large $S_i$) are located at the center, and the low degree nodes (low $S_i$) tend to the periphery. For the degree-driven $\mathbb{P}$ and $\mathbb{R}_1$ we observe a centralized information flow, in which the cross-talk between all nodes is primarily mediated by the hubs located at the core of the network (Fig. 4c, f, red). As predicted, the effect is more pronounced for the $\mathbb{P}$ dynamics, where $\omega$ is larger. Using the same network with

**Fig. 4** Universality classes of dynamic flow. We measured the flow $\mathcal{F}_i$ and $\mathcal{F}_{ij}$ through all nodes/edges for the six dynamic models of Table 1. We ran each dynamics on four different networks: Erdős-Rényi (ER, triangles), scale-free with scale-free weights (SF1, circles), scale-free unweighted (SF3, squares) and scale-free with normally distributed weights (SF4, diamonds). We compare the observed results with the predictions of (10) and (11) (solid lines), namely $\mathcal{F}_i$ vs. the weighted degree $S_i$ and $\mathcal{F}_{ij}$ vs. the product $S_iS_j$. **a, b** For $\mathbb{P}$ we predict $\omega = 5/3$ and $\xi = 2/3$ (solid lines), a degree driven flow, in agreement with the observed results (symbols). **c** To observe the implications of the degree driven flow we layout the nodes of SF1 with the hubs at the core and the low degree nodes at the periphery. As predicted, we find that the flow condenses around the hubs (large nodes, thick edges at center), which, in this dynamics, are responsible for most of the information flow throughout the network. **d–f** For $\mathbb{R}_1$ we predict $\omega = 2/3$, $\xi = -1/3$, again featuring hub-centralized flow. Here, the effect is weaker, due to the lower value of $\omega$ compared with $\mathbb{P}$. **g–l** For $\mathbb{E}$ and $\mathbb{B}$ we predict $\omega = 0$, $\xi = -1$, a homogeneous flow, where hubs and peripheral nodes have, on average, an equal contribution, as indicated in the layouts, in which flow is distributed evenly among all nodes/links. **m–r** For $\mathbb{M}$ and $\mathbb{R}_2$ we predict $\omega = -1$, representing degree-averting flow. In this class information flow favors the small nodes, avoiding the short paths that are centralized around the hubs, expressed in **o, r** by the dominance of the network periphery in both layouts. In all classes, the fact that the scaling ($\omega$, $\xi$) is independent of the network $A_{ij}$ (ER, SF1, SF3, SF4) confirms that our classification depends only on the system's dynamics

the same layout, the homogeneous $\mathbb{E}$ and $\mathbb{B}$ exhibit a non-centralized flow pattern, in which all nodes/pathways participate equally in spreading information (Fig. 4i, l, green). Finally, the degree-averting $\mathbb{M}$ and $\mathbb{R}_2$ show peripheral flow, in which information favors the longer, decentralized pathways that traverse through the exterior low degree nodes (Fig. 4o, r, blue).

Taken together, these distinct flow patterns, all obtained from the same network (SF1), illustrate the potential disparity between the static network topology and the actual dynamic pathways of information flow. Indeed, flow sometimes condenses around the hubs (red), distributes evenly across nodes (green), favors the network periphery (blue), or follows any other pattern within (6) and (9), as dictated by $\omega$ and $\xi$. Therefore to truly utilize networks as the tool they are designed to be—for visualizing the flow of information—one must use our analytically derived (6)–(11) to translate the network topology into actual pathways of information flow.

**Additional flow patterns**. At the heart of our analytical formalism lies Eq. (1), whose universal structure covers a broad range of steady-state dynamics, as captured in Table 1 and demonstrated in Figs. 2–4. To expand the applicability of our formalism, we now turn to two systems that extend beyond the boundaries of (1), and use numerical analysis to observe their flow patterns.

Epidemic spreading. The concept of dynamic flow can help us understand, and hence mitigate, the spread of epidemics, a most pertinent threat to our global health[46]. Indeed, to design efficient immunization strategies, we must identify the nodes with the highest contribution to the flow of information (or viruses). To observe this, we implement the susceptible-infected-recovered model, in which each node can be in one of three states, $S$, $I$, or $R$, representing a generalization of (1) to account for multidimensional activities $x_i(t)$. Freezing each node, we find that $\mathcal{F}_i \sim S_i$, representing a degree driven flow (Fig. 5a, red). This suggests that the optimal mitigation strategy is to immunize the hubs—a rather expected result. However, measuring the flow at later times, we find that the role of the hubs diminishes, indicated by the receding flow curve for large $S_i$ in Fig. 5b (green), up to a point where $\mathcal{F}_i$ sharply decreases with $S_i$, entering a rather extreme state of degree-averting flow (Fig. 5c, blue). Hence disease spreading exhibits an evolving flow pattern, being degree driven at the early stages of the contagion and degree averting as the system approaches the pandemic state. The reason for this transition is that the well-connected hubs become infected, and hence non-susceptible, at the early stages of the spread, at which point they cease to contribute to the viral flow (Fig. 5d).

To test these evolving flow patterns in an empirical setting, we used air-traffic data[46], capturing the international mobility of $7 \times 10^6$ individuals per day over the course of 3 years between $N = 1{,}292$ major international airports. Indeed, we find that flow evolves over time, condensing around different nodes at different stages of the contagion (Fig. 5e–j). These findings are crucial for developing efficient mitigation strategies based on air-traffic interventions, such as immunizing or quarantining passengers at

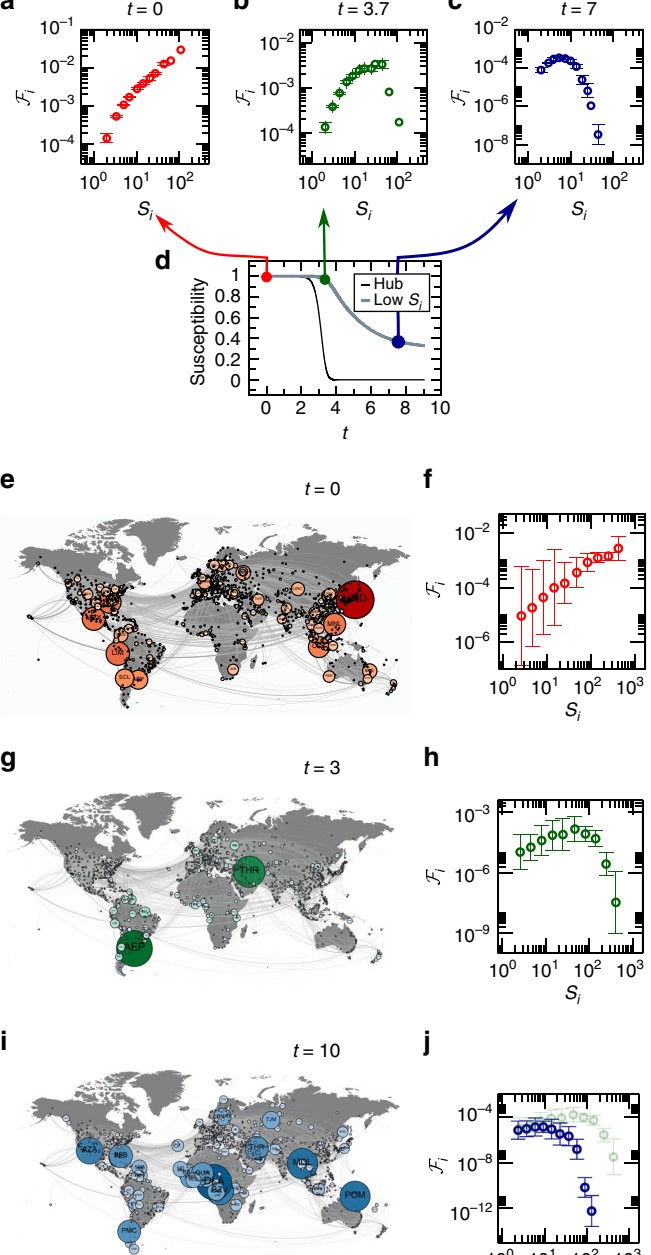

**Fig. 5** Evolving flow patterns in epidemic spread. We used the susceptible-infected-recovered (SIR) model to track the spread of disease in a weighted scale-free network and measured the flow through all nodes. The system exhibits evolving flow patterns: **a** $\mathcal{F}_i$ vs. $S_i$ at $t = 0$ exhibits a positive scaling, representing degree-driven flow (red). **b** At later times the role of the hubs gradually diminishes and $\mathcal{F}_i$ begins to decay in the limit of large $S_i$, a lack of scaling resembling homogeneous flow (green). **c** As the system approaches the pandemic state (large $t$) $\mathcal{F}_i$ begins to sharply decrease with $S_i$, entering a strongly degree-averting flow regime (blue). **d** Susceptibility vs. $t$ of a hub node (*black*) and a low degree node (*gray*). The hubs become infected (non susceptible) at earlier times, and hence cease to contribute to the spread—leading to the transition from degree-driven (red) to degree-averting (blue) flow patterns. **e** The flow through the empirical weighted international air-traffic network (nodes—international airports; edges—volume of human travel on route) under SIR, as represented by node size at $t = 0$, namely at the start of the outbreak. **f** $\mathcal{F}_i$ vs. $S_i$ for the aviation network at $t = 0$. The positive scaling confirms the degree driven flow. **g, h** At a later time we find, on the same network, a different flow pattern, in which the flow through the hubs begins to decline. **i, j** Finally, for large $t$ the flow enters the degree averting regime, as $\mathcal{F}_i$ strongly avoids the hubs. In **j** we show also the flow curve obtained at $t = 3$ (green watermark) for comparison. Indeed for $t = 10$ (blue) we observe a much stronger decline in hub-flow than that observed at $t = 3$, demonstrating the gradual evolution towards degree-averting flow. These evolving flow patterns illustrate the non-trivial mapping of the static topology to the observed dynamic behavior. See detailed description in Supplementary Note 5. Error bars represent 95% confidence intervals (Supplementary Note 3)

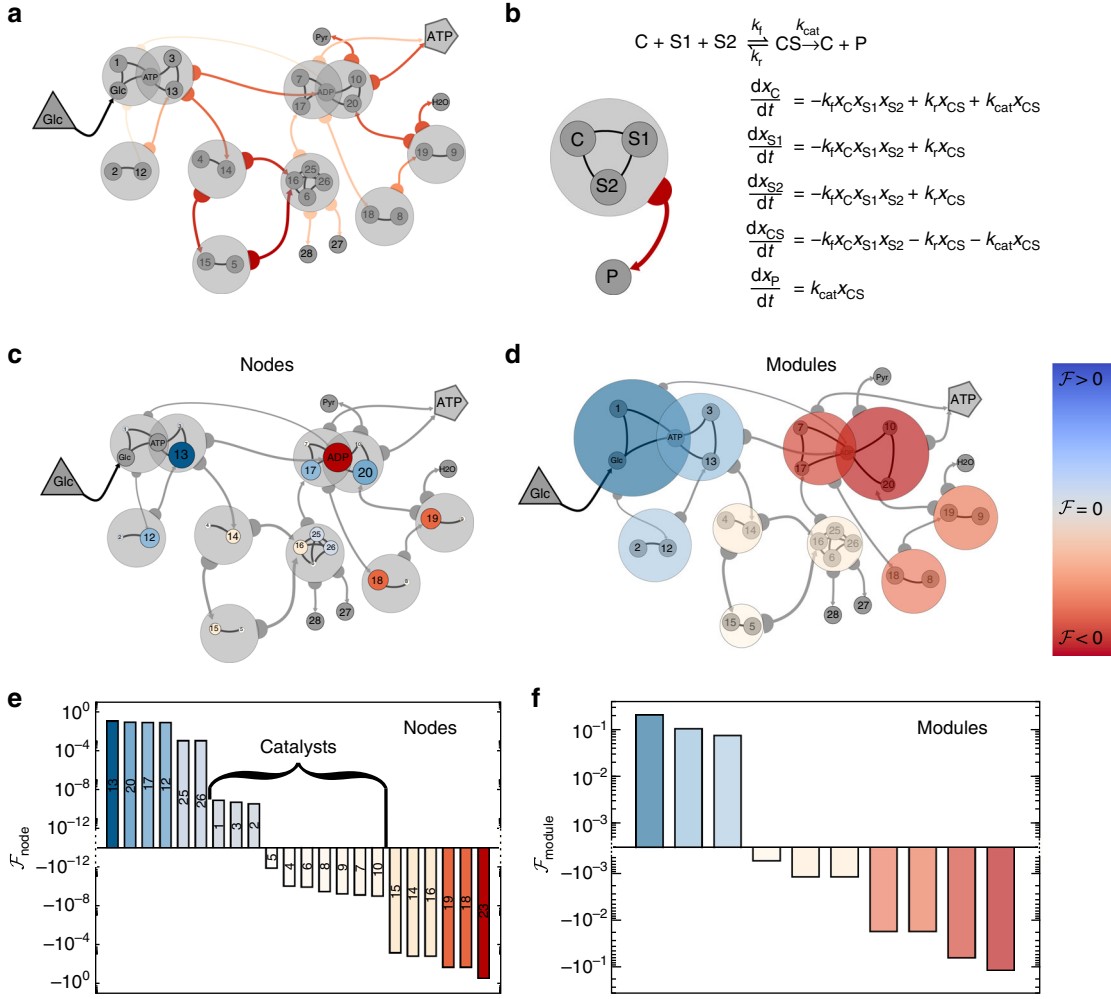

**Fig. 6** The balanced flow of a metabolic pathway. **a** The Glycolysis pathway, transforming glucose (triangle) into ATP (pentagon) visualized as a hyper-network. Each of the 10 modules (large gray circles) represents a chemical reaction, grouping together the substrates and catalysts (nodes) that participate in the interaction (catalysts $i = 1, ..., 10$; substrates $i = 11, ..., 28$). The arrows represent the flux levels from each interaction (module) to its product nodes. **b** Each interaction module comprises one or more substrates $S1,S2$, whose reaction, producing the product $P$, is catalyzed by the catalyst $C$. The bound catalyst $CS$ is an intermediate molecule, bridging between the interacting module ($C,S1,S2$) and its products ($P$). The dynamics of each interaction is captures through mass-action kinetic equations, mapping the flux emerging from each module (arrow) to the appropriate $n$th order equation term. **c** The flow from the input glucose to the output ATP through all nodes, as represented by node size. The pathway exhibits a balance of positive (*blue*) and negative (*red*) flows, representing the regulatory nature of metabolism, that restricts information flow from source (glucose) to target (ATP), ensuring a sustained level of ATP in the face of environmental perturbations. Catalysts have typically a tiny $\mathcal{F}_i$, as expressed by their small size. **d** The flow through each module (size/color of modules) represents the contribution of a reaction to information flow. **e, f** Bar plot capturing the observed flows through nodes/modules, portraying the balance of positive and negative flows. The small $\mathcal{F}_i$ associated with the catalysts is also marked. See detailed description in Supplementary Note 6

strategic air routes. Such interventions help reduce the spread of disease at the price of negatively impacting the mobility of people and goods, a burden, which may significantly impact the global economy. To minimize the damage, we seek optimal mitigation strategies, which employ minimal intervention. Our analysis suggests that hub-immunization, the commonly assumed strategy, is only effective at the early stages of the spread. As the spread unfolds the dynamic flow diffuses towards the peripheral pathways.

In a broader perspective, such time dependent flow patterns expose the limited predictive power offered by the static topology, which remains unchanged in time. Our formalism, on the other hand, was able to uncover the time evolving flow patterns, providing crucial insights on the dynamic nature of disease propagation, as well as practical implications on its mitigation. For the detailed analysis of this system see Supplementary Note 5.

**Metabolism.** As our final example we analyze information flow in Glycolysis (Fig. 6a[47]), a well-mapped metabolic pathway that consumes glucose (triangle) to form the energy-rich ATP molecule (pentagon). This biochemical sequence can be accurately modeled via mass-action-kinetics (Fig. 6b), giving rise to a rather rich module structure, including third and fourth order reactions, that help us extend our analysis beyond pairwise dynamics. Instead of $i,j$ links, we now have modules that represents chemical reactions, grouping together interacting substrates and catalysts (large gray circles), whose reactions generate flux (arrows), that link each module to its product molecules.

In this system, information flows from the input glucose to the output ATP, hence, by perturbing the glucose levels (a signal), we can measure the contribution of all reactants (nodes) or reactions (modules) to the flow, by sequentially freezing each node/module,

and tracking the consequent changes in ATP production (response). The resulting flow patterns, shown in Fig. 6c–f expose a balance of positive (blue) and negative (red) flows, indicating that although some nodes/modules enhance the spread, others mitigate it, by negatively contributing to the flow. This balanced picture illustrates the role of metabolism as a regulatory process, intended to sustain the desired output levels (ATP) in the face of environmental perturbations (glucose signal), achieved by restricting the efficiency of information flow. Interestingly, our flow analysis naturally distinguishes between substrates and catalysts, the latter showing extremely low $\mathcal{F}_i$ (Fig. 6c, e). This finding is supported by empirical observations, that biochemical outputs are highly insensitive to changes in enzyme concentration[48]. For the detailed analysis of this system see Supplementary Note 6.

## Discussion

From neuronal signals to gene regulation, complex networks function by enabling the flow of information between nodes. Understanding the rules that govern this flow is a crucial step toward establishing a theory of network dynamics. Our approach here is to separate the contribution of the topology ($A_{ij}$) from the dynamics ($\mathbf{M}, \omega, \xi$), allowing us to efficiently translate topological characteristics ($S_{i,\text{in/out}}, A_{ij}$) into dynamic predictions ($\mathcal{F}_i, \mathcal{F}_{ij}$). This will potentially enable us to leverage the vast amounts of data collected in recent years on the topology of real networks, into an understanding of their actual flow patterns. For instance, here we have shown that degree heterogeneity, a ubiquitous characteristic observed by almost all real networks[1], translates into one of three classes of flow: hubs may either dominate information flow (red), have no impact on the flow (green) or have a marginal role, effectively being the "shock-absorbers" of the network's signal propagation (blue).

Our derivations are exact for a random $A_{ij}$ with arbitrary degree/weight distributions, and under the assumption of small perturbations. We further establish their robustness when these assumptions are violated in Supplementary Note 4, confronting our predictions against large perturbations or non-random characteristics of $A_{ij}$, such as clustering $C$ and degree-correlations $Q$[49]. We find that extreme levels of $C$ or $Q$ may result in a systematic decrease in $\omega$, representing a reduction in the role of the hubs. This occurs due to the prevalence of loops in these limits, providing alternative pathways for the signals to bypass the well-connected nodes, a purely topological effect, observed independently of the dynamics. Still, even with these minor deviations in the precise values of $\omega$ or $\xi$, our macro-scale qualitative classification of flow patterns (degree-driven, homogeneous, degree-averting) remains unaffected, representing an intrinsic characteristic of the system's internal mechanisms $\mathbf{M}$, which is highly insensitive to microscopic discrepancies.

In a broader perspective, our predicted universality indicates that the macroscopic flow patterns of complex systems are controlled by only a few relevant parameters of the system's microscopic dynamics, in this case the leading powers of the expansion (7). Such disparity between the unlimited microscopic degrees of freedom, and the restricted set of macroscopic behaviors lays the basis for a statistical mechanics theory of network dynamics, allowing us to systematically translate a complex system's microscopic description, in terms of $A_{ij}$ and $\mathbf{M}$, to its anticipated large-scale dynamic behavior, e.g., centralized vs. peripheral flow.

## Methods

**Example: flow in regulatory dynamics**. Our formalism provides a step-by-step procedure to translate the topology $A_{ij}$ into dynamic flow $\mathcal{F}$, through the exponents $\omega$ and $\xi$. As an example we consider gene regulatory dynamics $\mathbb{R}$, where

(Supplementary Note 2)

$$\frac{dx_i}{dt} = -x_i^a + \sum_{j=1}^{N} A_{ij} \frac{x_j^h}{1 + x_j^h},$$ (12)

with $a, h > 0$. The contribution of all paths to the flow is governed by the exponents $\omega$ and $\xi$, which we now exemplify how to analytically extract in three steps: First, we break the dynamics into the three components of $\mathbf{M}$, providing

$$M_0(x) = -x^a, \quad M_1(x) = 1, \quad M_2(x) = \frac{x^h}{1 + x^h}.$$ (13)

We then construct the power series (7): first writing $W(x) = -M_1(x)/M_0(x) = x^{-a}$; then inverting it to obtain $W^{-1}(x) = x^{-1/a}$; finally, using (7) we construct the power series as

$$M_2(W^{-1}(x)) = \frac{x^{-\frac{h}{a}}}{1 + x^{-\frac{h}{a}}} = 1 - x^{\frac{h}{a}} + O\left(x^{\frac{2h}{a}}\right).$$ (14)

From (14) we extract the leading powers of $M_2(W^{-1}(x))$ as $\Gamma(0) = 0$ and $\Gamma(1) = h/a$. We next use $\Gamma(n)$ in (8) to predict $\omega$ and $\xi = \omega - 1$: here, since $\Gamma(0) = 0$, Eq. (8) predicts

$$\omega = 1 - \Gamma(1) = 1 - \frac{h}{a},$$ (15)

providing, for $\mathbb{R}_1$ ($a = 1$, $h = 1/3$), $\omega = 2/3$, a degree-driven dynamics, and for $\mathbb{R}_2$ ($a = 1$, $h = 2$) $\omega = -1$ a degree-averting dynamics. Hence, on the same network, a slight change in the dynamics (value of $h$) leads to fundamentally different flow patterns. A detailed analysis of all dynamics in Table 1 appears in Supplementary Note 2.

**Data availability**. We make our code DynamicFlow.m (Matlab) available with this submission. The code accepts a user defined network and dynamics and provides the dynamic flow patterns as output, together with the scaling relationships reported throughout the paper. Specifically, the code allows users to reproduce all results presented in the paper. All empirical networks used in this work are publicly available online.

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

## Author contributions

Both authors designed and carried out the research. U.H. conducted the derivations, data analysis, and numerical simulations. B.B. is the lead writer of the paper.

## Additional information

**Competing interests:** The authors declare no competing financial interests.

