## [Peer Review File · Nature Communications]

Reviewers' comments:

Reviewer #1 (Remarks to the Author):

The manuscript by U. Harush & B. Barzel titled "Dynamic patterns of information flow in complex networks" adopts a local perturbation approach with respect to a system's steady state to measure information flow in a class of network dynamical systems. Using the proposed measure, the paper reports findings of a diverse set of information flow patterns, demonstrating the important role played by dynamics in conjunction with network structure. I wish I could be more positive toward recommending the paper for publication in Nature Communications, but such potential recommendation is hindered by some critical issues.

1. (Lack of) novelty. The concept of local perturbation itself has been around for decades and commonly used in control theory and dynamical systems. The arguably novel aspect perhaps is the inclusion of network. However, this has already been studied in depth in some previous publications (Ref. [16]).

2. Ambiguity in some of the key results. The description of "freezing" the state of certain nodes, as described between Eq. (3) and Eq. (4), is particularly confusing. Suppose the state of node x_n is perturbed, what would be the mathematical definition of the response of the other nodes with versus without freezing the state of node x_i ? Without clarity, I found it challenging to appreciate much of the results that follow.

3. Restricted class of dynamics. Although it was argued that Eq. (1) represents a diverse set of commonly used dynamics, it was also mentioned that one key result in the paper is some analytical formula [Eq. (6)] that enables approximation of information flow. Is such formula a result of the particular choice of form of Eq. (1)? This is especially important to clarify as probably one of the most relevant results is the analysis of epidemic dynamics. However, as the authors noted themselves, such dynamics fall outside the scope of Eq. (1).

To summarize, in my view the manuscript in its current form lacks novelty and generality to worth publication in Nature Communications.

Reviewer #2 (Remarks to the Author):

I read with interest the paper “Dynamic patterns of information in complex networks” by Harush et al.

I found the paper technically correct, but I question the general importance of it. In the abstract the authors claim to have (developed) “a formalism (that) uncovers the universal rules that link structure and dynamic information flow in a broad range of non linear systems”.

I enclose here a list of questions

1) From eq. 1, authors limit the dynamics of the variable x of a given node i to two terms. The first is based on a generic function of the x_i itself, and the second on generic functions of the neighbours.

The first remark is that module structure is missing, i.e., if j and k (both neighbours of i) are connected or not does not make any difference. In principle this could hold in some cases (certainly not in flows) but in any case authors must stress more clearly that this is an approximation.

2) Probably in the same hypothesis of above authors introduce a linear response matrix in eq. 2. Also in this case this is not the more general situation one can have.

Authors then define the contribution of any single path to the flow of information from a given source. From that quantity, they first select a single node and a couple of nodes as path thereby studying (correctly) the flow through a specific node and through an edge (i.e., through the end vertices i, j of the edge).

3) Following the derivation in supplementary information, eq. 6 is valid under the following hypotheses (on top of the linear dynamic already stated)

- No degree-degree correlation

- Homogeneous weight distribution

- Large degrees only

This is analogous of typical Mean Field approximation in other field of physics, and it must be considered only as first approximation of phenomena.

4) I cannot sort out how the value of ω is computed. From eq. 1.54 in supplementary information it seems ω is extrapolated from data either from models or from experimental results. In any case, this holds only for large values of $S_{i,in}$. The collapse plot in Fig. 2 does not help since the data have been binned (and some of them show a rather large error bar).

The article would benefit from a table of different ω s for the various cases and from an explicit derivation at least in one of the cases presented.

5) at page 8 the authors refer to the dynamic backbone of information flow, but they do not define it

As minor remark, authors please check bibliography, I spotted the presence of many “Vespignany” instead of “Vespignani”.

Reviewer #3 (Remarks to the Author):

Dynamic patterns of information flow in complex networks

Uzi Harush, Barush Barzel

In this work the authors take a simple measure of information flow (how much does node j zig when node i zags) and use it to construct a metric of a path's importance to the global information flow. They then demonstrate that the importance of a node i to the global information flow scales with i 's in and out degree with an exponent determined by the form of the dynamics (at least for one broad

class of dynamics). They then give an expression for calculating this scaling exponent from the form of the dynamics. This set of derivations is repeated for the importance of an edge ij with similar results.

They then partition the dynamics into three classes based on the sign of this exponent (+, 0, -) and further demonstrate that the global importance metrics obey universal scaling behaviors. All this is backed up with numerous numerical studies and an analysis of an SIR model which is actually outside the model class of their derivation, and yet the metrics developed shed some new insight into the model's behavior.

Overall, I found this work to be well written, engaging, and scientifically sound. It is of a broad interest and is therefore appropriate for Nature Communication's diverse readership. I believe the results presented here can have a major impact on future works. I recommend it for publication in this journal without revision.

I will point out one issue which the authors may wish to address. The authors state that they are quantifying information flow, and while this is true in a broad sense, it is rather different than information as quantified by information theory which already plays a fairly major role in the study of dynamics on networks (e.g. transfer entropy and its kin). The authors may want to add a brief discussion about how what they quantify is information, though different than information as measured in bits. They could also draw analogies with chaos theory as their Gmn can be seen as somewhat akin to local Lyapunov exponents.

Reviewer #1

Comment:

The manuscript by U. Harush & B. Barzel titled “Dynamic patterns of information flow in complex networks” adopts a local perturbation approach with respect to a system’s steady state to measure information flow in a class of network dynamical systems. Using the proposed measure, the paper reports findings of a diverse set of information flow patterns, demonstrating the important role played by dynamics in conjunction with network structure.

Response:

We thank the Referee for this summary of our contribution and for his/her constructive comments that helped us improve our paper and its presentation. We agree that the important role of the interplay between dynamics and network structure is a core message of our paper, as is the notion that we can use local perturbations to observe how this interplay unfolds. We wish to emphasize, however, that this is not the *main* contribution of our work. Indeed, the main novelty of our proposed paper is that we can *analytically predict*, for a broad class of dynamic systems, *precisely how this interplay of network structure and dynamics will unfold*. Hence, our challenge is not just to *expose* the diverse set of information flow patterns, but rather to *derive* the universal rules that *predict* these patterns, showing analytically what pattern will be observed by which system.

Before we begin our detailed response to the rest of the Referee’s comments we wish to first provide the Referee with an overview of the significant revisions invested in the current paper. Some of these revisions were prompted by the Referee Reports, while others represent newly obtained data and added validation that we have acquired since the original submission. The result is, in our view, a significantly improved paper, with broader applicability and expanded validation. We hope that following our response/revisions, the Referee will find our paper suitable for publication in *Nature Communications*.

Overview of our main revisions and improvements:

Scope. We have broadened the class of dynamics on which we analyze the flow. First, we improve the implementation of the SIR model (see below), in which our flow analysis has already been shown to provide crucial insight. Additionally, we added an analysis of flow in metabolism, focusing on the well-established Glycolysis pathway, which comprises several third and fourth order interactions, therefore going beyond the pairwise structure of Eq. (1). Together, these two systems, the SIR model and metabolism, extend our analysis beyond its previous limits, showing the applicability of our conceptual framework on systems of increasing complexity. We emphasize again, that the core of our results remains the analytical solution for all systems in the form (1). To this we now added these numerical complements, to further demonstrate the insight enabled through our analysis.

Validity. In a newly added **Supplementary Section 4** we now extensively test the robustness of our predictions against common characteristics of a network’s fine structure, such as degree-degree correlations and clustering, that were neglected in the original submission. We also examine the validity of our predictions under large perturbations, going beyond the linear response approximation. The analysis helps further validate our predictions, showing that they are generally not sensitive to such discrepancies. Most importantly, when these features do lead to deviations from our predictions, our additional examination helps us understand their impact on the dynamic patterns of flow.

Presentation. We have now placed our results in context to better capture the theoretical concepts upon which we *build* (perturbative analysis) vs. those which represent our paper's *novelty* – the predicted universality classes of flow (following **Comment 1**). We thank the Referee for pointing us the fact that our original presentation may have been misleading in that sense. To further clarify the presentation we added a **Box** and a **Table** to the main text, where we exemplify the derivation of ω and ξ for a specific dynamics, and detail the obtained values of ω , ξ , and hence the dynamic universality classes, of all other analyzed systems. We also included an illustrative Figure (**Fig. 1**) to illustrate the meaning of *freezing* and *flow*, which the Referee indicated to be ambiguous (**Comment 2**).

Data. Since submission we were able to obtain improved data on the weighted global air-transportation network, allowing us to simulate the SIR model under realistic conditions. The results included in the current version of the paper (**Fig. 4**) continue to support the findings reported in the previous version, but under an improved empirical setting.

Below is our detailed point-by-point response to the Referee's comments:

Comment:

1. *(Lack of) novelty. The concept of local perturbation itself has been around for decades and commonly used in control theory and dynamical systems. The arguably novel aspect perhaps is the inclusion of network. However, this has already been studied in depth in some previous publications (Ref. [17]).*

Response:

We agree with the Referee that the notion of observing a system's dynamics through its response to perturbations is well-established, and, in fact, an integral part of the statistical physics paradigm. Its incorporation in the study of network dynamics was recently proposed in Ref. [17], as correctly indicated by the Referee, offering a framework for translating static structure (topology) into dynamic behavior (response to perturbation). Upon reading our introduction again, we believe that our choice of words in some cases may have indeed been inaccurate, leading to the impression that the *perturbative framework* represents our paper's novelty (see revisions below). However, we wish to emphasize that this is *not* the conceptual novelty of our contribution, but rather the *premise upon which we build*.

Relying on this premise - the well-established framework of perturbations - *our novelty* is that: (i) we introduce the measure of flow and (ii) we analytically derive the fundamental rules that govern its behavior. Hence our main contribution – indeed, *thee novelty of our paper* – is the analytical prediction that exposes how network structure, coupled with nonlinear dynamics, translates into information flow pathways. In other words – we agree that the idea of capturing network dynamics through perturbations is by now canonical. However, the *rules that govern* this propagation represent an open question, which we address here by identifying which nodes, links or pathways mainly support the diverse types of observed information flow.

Our analysis provides both a *quantitative* evaluation of the contribution of each node/path to the flow (Eqs. (6) - (11), **Fig. 2y,z**), as well as a *qualitative* understanding of the network's large scale flow patterns (**Fig. 3**). *Both of these results are currently unknown*. Indeed, at the current state-of-the-art, there is no systematic translation of static network structure into dynamic information flow. Certainly not one with the level of predictive power that we offer

here, as demonstrated in **Fig. 2**, where the diverse *zoo* of observed flow patterns, is shown to condense around two consistent universal flow functions.

Such analytical results at the intersection of complex networks and nonlinear dynamics are extremely rare, and yet highly pertinent. Hence, we believe, they represent a meaningful contribution to our currently lacking understanding of network dynamics.

We also wish to emphasize that the fact that perturbative theories are a well-established framework to analyze system dynamics is, in fact, a *strength* of our paper, showing that our predictions are not related to an obscure measure, tailored to our analytical derivations, but rather, to a commonly accepted method of observation, which has already been shown to be highly relevant in the context of network dynamics [Ref. 17]. Hence we agree that perturbation theory *has been around for decades*, but view this as precisely the foundation upon which we build: aiming to uncover a yet unknown universal prediction pertaining to a well-established concept that has been around for decades.

Revisions:

Upon careful reading of our introduction following this comment, we understand that our original wording was, indeed, misleading, pointing to our perturbative approach as the paper's perceived novelty, rather than to our analytical results that are derived via this approach. For instance, in the original introduction we wrote

*To observe these patterns we **develop** a perturbative approach...*,

representing a poor choice of words, as if the perturbative approach is being developed here. Hence we now replaced it with

*To observe these patterns we **employ** a perturbative approach, as commonly used to uncover information propagation in statistical physics systems ... We then analytically track the propagation of signals between nodes...*

We also include the relevant citations to clarify that perturbations are the foundations upon which we build, aiming at our true novelty: *analytically tracking the propagation of signals*. We believe that this and other similar edits that we introduced where appropriate will lead to a clearer presentation, highlighting our novelty, and, on the other hand, better accrediting the well-established framework that we used to obtain it.

Comment:

- 2. Ambiguity in some of the key results. The description of "freezing" the state of certain nodes, as described between Eq. (3) and Eq. (4), is particularly confusing. Suppose the state of node x_n is perturbed, what would be the mathematical definition of the response of the other nodes with versus without freezing the state of node x_i ? Without clarity, I found it challenging to appreciate much of the results that follow.*

Response:

We find this to be a crucial comment and thank the Referee for prompting us to clarify the concept of *freezing* and its implications on the flow. To understand this concept, consider first a one-dimensional network, including a sequence of nodes s, i and t (source, intermediate and target) as shown in **Fig. 1a**. At the steady state their activities are given by x_s, x_i and x_t . Next we introduce a perturbation on the source s , shifting its state to $x_s + dx_s$ (vertical arrow), and track the propagation of this perturbation along the linear path $s \rightarrow i \rightarrow t$. This propagation is manifested through a series of responses, in which the steady state of all nodes is sequentially perturbed: x_s is changed to $x_s + dx_s$ and as a result x_i changes to $x_i + dx_i$, finally leading to x_t being perturbed to $x_t + dx_t$, hence dx_t represents the magnitude of *information* that propagated from the source s to the target t . Next we wish to evaluate how much of that information flowed through the intermediate node i , providing us with \mathcal{F}_i , the flow from s to t through i . To evaluate this flow we wish to eliminate i 's contribution to the propagation, requiring us to forcefully set dx_i to zero, thus terminating all of the information that was transferred through that node. This is precisely the meaning of *freezing* node i , namely preserving its unperturbed state x_i , therefore blocking its ability to propagate the perturbation. In this case, such freezing completely terminates the propagation, resulting in $dx_t^{\{i\}} = 0$ ($dx_t^{\{i\}}$ represents t 's response under the freezing of i). Therefore i 's contribution to the flow is 100%, namely *all* information from s to t flows through i (**Fig. 1b**), a trivial consequence of the linear topology that we used in this case.

Note that *freezing* i is different from *removing* it, a process that changes the network structure itself, and will therefore result in a subsequent change in the steady states x_s, x_i, x_t of *all* three nodes, regardless of the perturbation. It is also different from setting $x_i = 0$, an intervention on i 's steady state, which, in practice, introduces a *new* perturbation that will result in a significant response of both s and t , unrelated to the propagation from s to t . Hence freezing is a more subtle intervention - eliminating only the *change* dx_i through which i enables information to flow.

To illustrate flow in a more complicated scenario consider now the network of **Fig. 1c**, where s can be linked to t through two potential pathways. Here freezing i will have little impact on the flow through the top path, therefore $dx_t^{\{i\}}$ will be reduced to half of its original response dx_t , and hence \mathcal{F}_i in this case is only 50%, indicating that now i transmits only half of the information flow from s to t . In a

Figure 1. Dynamic freezing and flow. (a) Information, in the form of an activity perturbation dx_s (vertical arrow) propagates from the source s to the target t through the intermediate i , by causing the middle perturbation dx_i . (b) Freezing i terminates the intermediate response, setting $dx_i = 0$. As a result no information reaches t , hence i contributes 100% of the information flow from s to t . (c) - (d) Here freezing i still allows information flow through the upper pathway, hence $\mathcal{F}_i = 1/2$. (e) - (f) In a complex network environment the contribution of i is difficult to extract, illustrating the challenge that we address in the current paper.

complex network (**Fig. 1e,f**), the contribution of each node to the flow becomes a significantly more challenging problem – *which is precisely what we address in our paper*.

Revisions. We have now, following this comment, added an illustrative Figure (**Fig. 1**) to the main text visualizing the concept of *flow* and *freezing* along the lines of the above explanation.

Comment:

3. *Restricted class of dynamics. Although it was argued that Eq. (1) represents a diverse set of commonly used dynamics, it was also mentioned that one key result in the paper is some analytical formula [Eq. (6)] that enables approximation of information flow. Is such formula a result of the particular choice of form of Eq. (1)? This is especially important to clarify as probably one of the most relevant results is the analysis of epidemic dynamics. However, as the authors noted themselves, such dynamics fall outside the scope of Eq. (1).*

Response:

This Comment, in our opinion, touches on one of the central aspects of our contribution, the fact that we seek a general formalism that can treat diverse dynamical systems (and let us clarify that by *diverse* we mean a *broad scope* of dynamics, but clearly not *all* of dynamics). We now understand that in our original presentation the limits between our analytical predictions and our complementing numerical analyses were not clearly marked, hence the general applicability, but also the specific restrictions, accompanying our analytical predictions remained somewhat vague. The Comment led us to clarify these aspects of our presentation, and also prompted us to further expand the applicability of our conceptual framework, by adding an additional system (metabolic), which goes beyond the pairwise dynamics covered by our Eq. (1). Therefore, we wish to thank the Referee for the opportunity to clarify and expand the scope of our predictions.

Before we discuss our specific revisions, let us first address the Referee’s specific questions:

- Equations (6) - (8) and later (9) – (11), which predict the flow through nodes and edges, are an analytical result obtained for *all* dynamic models within the form of Eq. (1). *Hence, they are not the result of a particular choice within (1)*. The specific form of Eq. (1), *e.g.*, SIS or ecological dynamics, is encapsulated within the value of the universal exponents ω and ξ in (6) and (8), which we derive directly from the nonlinear functions $M_0(x)$, $M_1(x)$ and $M_2(x)$ of Eq. (1). Therefore the extreme diversity of flow patterns that we find across different dynamical systems (**Fig. 2a - x**) is solely rooted in the values of these exponents – highlighting their importance in predicting the system’s dynamic flow. In Eqs. (7) and (8) we show how to extract these two exponents directly from (1) and in **Supplementary Section 2** we explicitly present the actual derivation for each of the six analyzed models. To summarize, all our results, including **Figs. 2** and **3** are analytically derived from (1), not relying on a specific choice within the bounds of this equation.
- Our final analysis of the SIR model (**Fig. 4**), and our newly added analysis of metabolism (**Fig. 5**) are, as the Referee correctly identifies, beyond the scope of Eq. (1). In these systems the patterns of flow were extracted numerically. We deliberately include systems that are not within our analytical framework, to test the applicability of our formalism under broader conditions. The idea is to show that our formalism can

provide insight, even outside the bounds of the analytically soluble (1). The SIR model represents an especially relevant application, as the flow (of viruses) sheds direct light on immunization strategies – a pertinent challenge in our globally connected society. Our results, showing that flow evolves as the epidemic spreading unfolds, provide important insight and practical implications on immunization – *insight that was only enabled thanks to our newly introduced perspective on information flow.*

To summarize, the concept of flow applies in diverse contexts – as demonstrated here on *eight* different systems. Many of these systems (*six* included in our analysis) can be cast in the form of Eq. (1), and hence treated analytically, others (*two* – SIR and metabolism) can only be analyzed numerically. We find it important to cover both ranges, especially to show that the insight provided by our analytical derivations sheds light also on systems that are beyond the analytically tractable (1).

Generality of our analytical results. Our analytical results cover *all* dynamics that can be expressed in the form of Eq. (1), a rather general family of dynamics, given the freedom to select the nonlinear $M_0(x)$, $M_1(x)$ and $M_2(x)$, enabling one to construct practically *all forms of pairwise dynamics*. We exemplify this in the paper by demonstrating our predictions on *six* different realizations of Eq. (1), from areas as diverse as social, biological and ecological systems (**Table 1**). As emphasized above, such broad scope seldom succumbs to analytical treatment, especially under the challenging combination of complex networks and nonlinear dynamics, and hence we believe our analytical results mark a significant advance towards systematically mapping structure (*e.g.*, degrees, pathways) to dynamic behavior (*i.e.* flows).

Revisions. We have now, following this remark, clarified the scope of our results, emphasizing which of them is obtained analytically, and under what conditions, and which are a result of numerical analysis. We also added specific notes in the Supplementary Information (Grey boxes), where we explicitly discuss our model assumptions and approximations, and hence the restrictions on our analytical findings.

To further demonstrate the applicability and potential insight of our formalism we added a flow analysis of the Glycolysis metabolic pathway, representing one of the most accurately mapped sub-networks in the biological domain. Together, in its current form, our paper now includes six diverse systems that are all analytically tractable within the framework of Eq. (1), and two additional system (SIR, metabolism) that extend the validity of our analysis beyond Eq. (1).

Finally, we added a Supplementary Section (4), where we test our predictions under challenging conditions, such as large perturbations, and common topological features, specifically clustering and degree-correlations. Such additional numerical validation helps us further support the generality and robustness (as well as limitations) of our theory.

Comment:

To summarize, in my view the manuscript in its current form lacks novelty and generality to worth publication in Nature Communications.

Response:

We hope that following our response the Referee will find our thoroughly revised manuscript suitable for publication in *Nature Communications*.

Reviewer #2

First and foremost, we wish to thank Referee 2 for his/her thorough reading of our paper. The constructive and thoughtful comments provided in the Report led us to significantly improve our presentation and add additional layers of validation and applicability, beyond the originally presented results. Our detailed response to the Referee's comments appears below, but first, let us summarize the main improvements to the current version of the paper.

Applicability. We added the analysis of the Glycolysis metabolic network (**Fig. 5**), to demonstrate our treatment of flow in networks with module structure (following **Comment 1**). This pathway includes second, third and fourth order biochemical reactions, going beyond the limit of pairwise interactions that are covered within our Eq. (1).

Validation. In a newly added Supplementary Section (**4**) we extensively test the robustness of our main predictions against commonly observed features of a network's fine structure, specifically degree-degree correlations and clustering (following **Comment 4**). We also examine the flow under large perturbations, testing our predictive power in the limit where the linear response framework is superseded (following **Comment 2**). The results help us support the validity of our predictions, and, at the same time, better characterize their boundaries and their required corrections when clustering/degree correlations become dominant. We wish to thank the Referee for pushing us to examine these effects.

Presentation. To clarify the presentation we added a **Box** and a **Table** to the main text, where we exemplify the derivation of ω and ξ for a specific dynamics, and detail the obtained values of ω, ξ for all analyzed systems (following **Comment 6**). We also break down the collapse plot of **Fig. 2**, to allow a better assessment of the accuracy of our theoretical prediction and to track the precise behavior of the flow under different limits, for instance pointing to where $S_{i,in/out}$ is large/small (following **Comment 5**). Finally, we make our assumptions (random network, small perturbations) more explicit, in the text, in the Supplementary Information (Grey Boxes) and in the caption of the illustrative Fig. 1.

Data. Since submission we were able to obtain improved data on the weighted global air-transportation network, allowing us to simulate the SIR model under highly realistic conditions. The results included in the current version of the paper (**Fig. 4**) continue to support the findings reported in the previous version, but under a significantly improved empirical setting.

Below is our detailed point-by-point response to each of the Referee's comments:

Comment:

I read with interest the paper “Dynamic patterns of information in complex networks” by Harush et al.

I found the paper technically correct, but I question the general importance of it. In the abstract the authors claim to have (developed) “a formalism (that) uncovers the universal rules that link structure and dynamic information flow in a broad range of nonlinear systems”.

Response:

We wish to thank the Referee for this feedback. Indeed, the general applicability of our formalism is an important component of our contribution, and we thank the Referee for prompting us to significantly strengthen this aspect of our paper.

Comment:

I enclose here a list of questions:

- 1. From eq. (1), authors limit the dynamics of the variable x of a given node i to two terms. The first is based on a generic function of the x_i itself, and the second on generic functions of the neighbours. The first remark is that module structure is missing, i.e., if j and k (both neighbours of i) are connected or not does not make any difference. In principle this could hold in some cases (certainly not in flows) but in any case authors must stress more clearly that this is an approximation.*

Response:

The Referee correctly points out that our analysis focuses on *pairwise interactions*, as also implied by the notion of an interaction *network* (A_{ij}), in which links connect at most a pair of nodes. This is by no means a narrow class of dynamics, as demonstrated by the broad range of systems that we include in our analysis, from epidemics to biochemical interactions. We agree, however, that adding higher order interactions – or *module structure* – constitutes a meaningful expansion of our current results. This requires us to generalize from a standard *network* to a *hyper-graph* with hyper-links that connect more than just node pairs, but rather *modules* comprising several nodes each, thus enabling higher order interactions. In principle, observing the flow in such hyper-networks can be achieved by following the precise procedure outlined for regular pairwise interactions: (i) introducing perturbations to the system; (ii) freezing certain nodes/paths and (iii) observing the system’s response. Of course, now freezing an *edge*, should be properly replaced with freezing a *module* – a process we now explain and demonstrate in the current version of the paper. The results are just as insightful, exposing the nodes/modules that contribute most to the transfer of information in the system.

Revisions:

Following this comment we explicitly stress that our analytical results are focused on pairwise dynamics, as suggested by the Referee, excluding module structure.

However, prompted by this comment we have decided to expand our examination of flow beyond pairwise dynamics, and observe the influence of module structure. Therefore, we have now added an additional system to our analysis – the Glycolysis metabolic pathway, one of the most fundamental and accurately mapped biological mechanisms that transform glucose to form the *energy molecule*, ATP. This biochemical circuit comprises a set of second to fourth order reactions (*i.e.* modules), whose dynamics can be accurately simulated using mass-action-kinetics, thus providing a highly validated and reliable system on which to examine dynamic flow.

This newly added analysis is presented in the new **Fig. 5** and detailed in **Supplementary Section 6**, where we explain specifically how to measure the flow through a node and through a module. Interestingly, we find that the Glycolysis pathway exhibits a balance of positive and negative flow contributions, and hence, when intact, is designed to mitigate information flow, as one module balances out the response of another. This is perhaps an indication of the biological role of this metabolic pathway as a *regulatory* system, whose main contribution is to control the effect of environmental perturbations, by countering information flow from source (glucose) to target (ATP). Indeed, if information flow becomes too efficient, local or external perturbations may lead the system to dramatic response, which is often undesirable for a biological system.

We believe that this newly added system provides a meaningful contribution, strengthening our paper and expanding its breath, and we therefore wish to thank the Referee for prompting us to push forward the frontiers of our work.

Comment:

2. *Probably in the same hypothesis of above authors introduce a linear response matrix in eq. (2). Also in this case this is not the more general situation one can have.*

Response:

The linear approximation that we employ is a most common tool to analyze nonlinear dynamics, in the majority of cases - the *only* tool that allows for systematic analytical treatment. Our analysis, and hence all our predictions that follow, is exact in the limit of small perturbations, which is precisely the focus of our work. On the other hand, we agree with the Referee that it is important to examine the applicability of our results under more general situations – for instance, in the case of intermediate or even large perturbations. Therefore, prompted by this comment, we decided to test our predictions under this limit of increasing perturbation size, gradually departing from the linear response regime. We now added a section to the Supplementary Information (**4**) where we test our predicted flow exponents (ω, ξ) for a range of perturbation sizes, from 10% to 80%. Such large signals test our linear response approximation significantly beyond its theoretical range of validity. Strikingly, we find that our predicted exponents remain accurate even under these challenging conditions, indicating the robustness of our predictions also under large perturbations.

Such results, while surprising at first glance, are in fact an expected consequence of our approach to base the observed universality classes of flow on the scaling exponents ω and ξ . Indeed, scaling relationships are known to exhibit a high level of robustness, determined by only a small number of relevant parameters and at the same time being highly insensitive to minor discrepancies in the model assumptions. This is the reason that the classic

universality observed in critical phenomena is also formulated in terms of critical exponents, and, in fact, also represents our current motivation to focus on scaling relationships – knowing that their values are highly insensitive to the abstractions of the analytical model.

In a broader perspective, the insight provided by our predictions is not limited just to the specific values of ω or ξ . Indeed, a meaningful aspect of our contribution is that it sheds light on the large scale behavior of the network. For instance, $\omega = 1$ or 2 indicates that hubs dominate information flow – a macroscopic feature that drives the system’s behavior. On the other hand, $\omega = -1$ or -2 implies the exact opposite, that hubs behave as shock absorbers, having a negligible contribution to the flow. Such macro-level dynamic features are an intrinsic characteristic of the system, engrained in its driving mechanisms, hence extremely unlikely to change just because the signal size is increased. Therefore, even if some of our model abstractions (*e.g.*, small perturbations) are violated, this may, at most, cause subtle inaccuracies in the *precise* values of ω, ξ , but cannot, in the vast majority of cases, transform the system from, say, degree-driven to degree-averting.

Revisions:

We added a new Supplementary Section (4) where we extensively test our predictions under large perturbations, up to 80%. We find that not only do our qualitative predictions hold, but even our quantitative predictions, *i.e.* the precise value of ω remain valid. Our predictions do fail when the perturbation reaches a level where $dx \gg x$, namely when the perturbation changes the node’s activity by an order of magnitude (or more). Under such extreme signal size our linear response analysis, indeed, begins to fail. Still, as long as the signal size is of the same order as the node’s activity (*e.g.*, 80%), our predictions remain valid.

Comment:

3. *Authors then define the contribution of any single path to the flow of information from a given source. From that quantity, they first select a single node and a couple of nodes as path thereby studying (correctly) the flow through a specific node and through an edge (i.e., through the end vertices i, j of the edge).*

Response:

Indeed, a concise summary of our analytical results.

Comment:

4. *Following the derivation in supplementary information, eq. 6 is valid under the following hypotheses (on top of the linear dynamic already stated)*
 - i. *No degree-degree correlation*
 - ii. *Homogeneous weight distribution*
 - iii. *Large degrees only*

This is analogous of typical Mean Field approximation in other field of physics, and it must be considered only as first approximation of phenomena.

Response:

Our analytical predictions (Eqs. (6) - (11)) rely on *two* of the (three) assumptions mentioned by the Referee:

- ***Little degree-degree correlations*** – an approximation that we now thoroughly examine.
- ***Large weighted degrees*** – representing a most relevant asymptotic limit in the frequently encountered scale-free regime.

We wish to clarify that our derivation *does not* assume a homogeneous weight distribution (assumption (ii) in Comment). To the contrary, while our results do not depend on it, they are most relevant, significant and insightful under the conditions of extreme heterogeneity, *e.g.*, a scale-free network with scale-free weights. Such heterogeneity in degrees/weights, indeed, represents a most common characteristic observed by a large number of real world networks, including both the model (*e.g.*, SF1) and real (PPI, Eco, UCInonline, Epoch) networks that we analyze in the paper.

As the Referee correctly points out, our two assumptions represent a mean-field approach, however one that is profoundly distinct from the naïve notion of *mean-field*. Indeed, mean-field calculations often reduce the system to a single *average* node, by that smoothing out *all* inhomogeneity between the nodes. Here we take a much more subtle approximation, known as the *degree-based mean-field* (DBMF) approach [*Rev. Mod. Phys.* **87**, 925 (2015)], which applies mean-field not to the *nodes*, but rather to their *neighborhoods*.

To understand this consider a pair of nodes, a hub i and a peripheral node j with degrees $k_i \gg k_j$. These nodes are highly distinct, and hence cannot be averaged without losing crucial information about the system's heterogeneity. The point is that in the DBMF approach we avoid such crude averaging. Instead, DBMF assumes that even if i and j are different, their *neighborhoods* are similar, namely that the k_i neighbors of i are sampled from the same distribution as the k_j neighbors of j . As a result i 's average neighbor has similar characteristics to those of j 's average neighbor, even though i has many more neighbors than j . Therefore DBMF preserves all aspects of the system's dynamics that are driven by the network's heterogeneity (*i.e.* that i and j are extremely different). At the same time it overlooks the subtle effects driven by microscopic correlations between a node and its immediate environment (*i.e.* that i and j 's neighbors are different). Such fine-structural characteristics, including, for instance, degree-degree correlations, have, typically, only a nuanced effect on the system's macro-level behavior.

Validity of the DBMF approximation. The DBMF approximation is a common step in analyzing network dynamics, mostly used in the context of epidemic spreading (SIS, SIR models), providing highly reliable results even under empirically observed levels of degree correlations. The results we obtain here are no exception. Indeed, the many empirical networks that we analyze in the paper, from biology, sociology and ecology, all feature some level of degree correlations, in some cases up to $Q = -0.2234$ (Human PPI). Still, both our qualitative and quantitative predictions seem to be robust against these discrepancies. This robustness is reminiscent of the one we report above in the case of large perturbations (**Comment 2**). As before, it is a consequence of the insensitivity of scaling exponents to microscopic discrepancies in the model approximation.

Yet, following this Comment we decided to go further and systematically test the impact of deviations from DBMF, by testing our predictions against two common relevant network

characteristics: *degree correlations* (Q), as mentioned by the Referee, and *clustering* (C), an additional feature present in many real networks. Hence, we generated a series of scale-free networks with increasing levels of Q and C , and tested our predictions against them. As expected, we find that our results are sustained even for relatively large values of Q and C . The slight deviations that we observe in the case of large Q, C indicate a systematic decline in the role of the hubs, a consequence of the prevalence of loops that allows for alternative flow pathways around the hubs in these regimes. The detailed analysis appears in the newly added **Supplementary Section 4**. As in previous comments, we wish to thank the Referee for motivating us to include these additional tests, which, indeed, proved insightful, and helped us gain a better understanding of the impact of Q and C on our predicted patterns of flow.

The limit of large (weighted) degrees. The Referee is correct - our results, which are based on a leading order approximation in S_i^{-1} , are exact in the asymptotic limit of large S_i , (small S_i^{-1}). Therefore, one may observe some deviations for small S_i . This approximation is motivated by the ubiquitous scale-free phenomena, for which the asymptotic regime of large S_i represents a highly relevant limit. Indeed, in a scale-free environment the majority of pathways pass through the hubs, and hence their role in the dynamic flow of information, helps us illustrate the overall flow patterns in the network. This is clearly observed in **Fig. 3m - r**, where the large-scale flow patterns – *centralized, homogeneous or peripheral*, are fully predicted by the role of the hubs – *degree-driven* ($\omega > 0$, red), *distributed* ($\omega = 0$, green) or *degree-averting* ($\omega < 0$, blue).

Quantitative vs. qualitative accuracy. Finally, we note that our predictions include two levels: *quantitative* predictions on the precise value of ω and ξ , and *qualitative* predictions on the large scale patterns of flow – centralized, homogeneous or peripheral (**Fig. 3**). Our formalism allows us to predict both levels of behavior directly from the functional form of the dynamics (**M**), based on some abstractions, such as the assumption of small perturbations or the absence of significant degree-correlations (Q). The important point is that even if, for some dynamic systems, our *quantitative* predictions are sensitive to, *e.g.*, a large Q , the *qualitative* predictions will remain robust. To understand this, consider our finding that extreme Q values led to a decline in the role of the hubs. The meaning of this is that if, for some system, we predict $\omega = 1$, we may find that the actual observed value is slightly lower, say $\omega \approx 0.8$ or 0.9 . This deviation is certainly important. Still, it does not harm the *qualitative* prediction that the flow in this system is degree-driven ($\omega > 0$), and hence centralized along the hubs, a meaningful prediction in and of itself, that is *only enabled thanks to our formalism*. Therefore, even if the exact value of ω and ξ may, at times, be sensitive to our specific model assumptions, the *dynamic class* of each system (centralized, homogeneous, peripheral) is highly robust.

Revisions. Following this comment we made several revisions to the current manuscript to clarify our model assumptions and to further validate the broad applicability of our theoretical predictions:

- **Transparent derivation.** We now discuss in detail, during our derivation in **Supplementary Section 1** our precise assumptions and approximations, specifically the notion of the DBMF approach. The main assumptions and important caveats or conclusion are now highlighted in a Grey Boxes in the appropriate sections.
- **Further validation.** To address the empirical relevance of our DBMF based derivations, we now systematically examine the impact of fine-structure on the validity of our theoretical results. We focus on two notable meso-scopic topological characteristics, which violate the DBMF approximation: *degree-degree correlations* (Q), which were

cited by the Referee, and *clustering* (C), which tends to zero in random networks, but is often higher in real networks. The results, discussed above, are presented and analyzed in the newly added **Supplementary Section 4**.

- **Discussion.** We add a brief discussion in the Summary section of the main paper on the validity and limitations of our theoretical predictions, referencing the appropriate locations in the Supplementary Information.

Comment:

5. *I cannot sort out how the value of ω is computed. From eq. 1.54 in supplementary information it seems ω is extrapolated from data either from models or from experimental results. In any case, this holds only for large values of $S_{i,in}$. The collapse plot in Fig. 2 does not help since the data have been binned (and some of them show a rather large error bar).*

Response:

It seems, for this Comment, that our original presentation may have not been clear enough regarding the derivation of ω . We have made several changes to correct for this, however, let us first address the Referee’s specific question:

The value of the exponents ω and ξ is an *analytical result*, derived directly from the system’s intrinsic dynamics. It is *not* extracted from numerical or empirical data, but rather *predicted* from the structure of the system’s dynamic equation (Eq. (1) in main text). To obtain these exponents we first separate the nonlinear functions comprising Eq. (1), obtaining the three functions $M_0(x)$, $M_1(x)$ and $M_2(x)$. These functions describe the internal dynamics of the self and pairwise interactions of the system’s components - capturing the *physics* of the system - independent of the network A_{ij} . From these three functions we construct $W(x) = -M_1(x)/M_0(x)$, and then invert it to obtain the function $W^{-1}(x)$. Finally, we introduce this inverted function as input to $M_2(x)$, and expand it as a power series

$$M_2(W^{-1}(x)) = \sum_{\Gamma(n)} C_n x^{\Gamma(n)}.$$

The value of ω is then determined from the leading powers in the above expansion, $\Gamma(0)$ and $\Gamma(1)$, as shown in Eq. (8) in the main text. Hence ω (and consequently $\xi = \omega - 1$) are analytically derived from the system’s nonlinear dynamics $M_0(x)$, $M_1(x)$ and $M_2(x)$.

This step-by-step procedure, which may seem a bit *magical* at first glance, is a result of the rather elaborate derivation presented in **Supplementary Section 1**. Its idea is to systematically take a system’s dynamics \mathbf{M} and translate it, via a set of well-defined mathematical steps - division, inversion, expansion, in into the predictive exponent ω .

Revisions:

We believe the newly added **Box I** (following **Comment 6**), in which we explicitly demonstrate the above calculation for a specific dynamic model will help clarify the precise origins of our predicted exponents ω and ξ . We also, in the updated submission, explain this in detail directly after Eq. (1.55) in **Supplementary Section 1**, including a Grey Box to highlight the important points. Finally, we expanded **Supplementary Section 2**, in which we present the precise derivation of these exponents for *all* models analyzed in the paper.

Comment:

6. *The article would benefit from a table of different ω s for the various cases and from an explicit derivation at least in one of the cases presented.*

Response:

We agree. We now added a table (**Table I**) that summarizes our models and their classification, and a box (**Box I**), which illustrates the calculation of ω and ξ for a specific model, as suggested by the Referee.

Comment:

7. *At page 8 the authors refer to the dynamic backbone of information flow, but they do not define it.*

Response:

We agree - the dynamic network backbone was only loosely defined in the previous version, and we wish to thank the Referee for pointing this out to us, and prompting us to use more careful terminology. Our goal was not to define a *quantitative* measure but rather expose the *qualitative* distinction between the possible flow patterns: *centralized*, in which the flow is concentrated within a small number of top ranking components around the hubs, *homogeneous*, in which it is evenly dispersed throughout the network, and *peripheral*, in which the greater part of the flow streams through the small nodes. These qualitative behaviors are clearly observable in **Fig. 3m - r**, where node/link size scales with its contribution to the flow. Hence, the different classes are characterized by visible qualitatively distinct macroscopic flow patterns.

Revisions. To avoid confusion in the current version we refrain from using the ill-defined *backbone* terminology, instead we highlight the qualitative large scale differences in the main flow pathways, as visualized in **Fig. 3**.

Comment:

8. *As minor remark, authors please check bibliography, I spotted the presence of many "Vespignany" instead of "Vespignani".*

Response:

Corrected.

Reviewer #3

Dynamic patterns of information flow in complex networks

Uzi Harush, Barush Barzel

In this work the authors take a simple measure of information flow (how much does node j zig when node i zags) and use it to construct a metric of a path's importance to the global information flow. They then demonstrate that the importance of a node i to the global information flow scales with i 's in and out degree with an exponent determined by the form of the dynamics (at least for one broad class of dynamics). They then give an expression for calculating this scaling exponent from the form of the dynamics. This set of derivations is repeated for the importance of an edge ij with similar results.

They then partition the dynamics into three classes based on the sign of this exponent (+, 0, -) and further demonstrate that the global importance metrics obey universal scaling behaviors. All this is backed up with numerous numerical studies and an analysis of an SIR model which is actually outside the model class of their derivation, and yet the metrics developed shed some new insight into the model's behavior.

Overall, I found this work to be well written, engaging, and scientifically sound. It is of a broad interest and is therefore appropriate for Nature Communication's diverse readership. I believe the results presented here can have a major impact on future works. I recommend it for publication in this journal without revision.

Response:

We wish to thank the Referee for this precise and highly appreciative summary of our contribution, capturing in a concise fashion the main highlights of our work.

Comment:

I will point out one issue which the authors may wish to address. The authors state that they are quantifying information flow, and while this is true in a broad sense, it is rather different than information as quantified by information theory which already plays a fairly major role in the study of dynamics on networks (e.g. transfer entropy and its kin). They authors may want to add a brief discussion about how what they quantify is information, though different than information as measured in bits. They could also draw analogies with chaos theory as their G_{mn} can be seen as somewhat akin to local Lyapunov exponents.

Response:

We thank the Referee for these insightful observations, which help us better place our terminology and results in context. Our definition of information flow is indeed different from the entropy-based notion of information transfer, and we have now clarified this distinction in the illustrative **Fig. 1** of our revised paper.

The Referee also correctly points out the link between our G_{mn} and the linear stability analysis, from which Lyapunov exponents are extracted. The main difference is in the form of the perturbation that we introduce into the system: in the Lyapunov framework, the system is nudged at $t = 0$ from its current state to a slightly perturbed state. As a result its

trajectory in phase-space is shifted, where, roughly speaking, it can either diverge to a completely new path (chaos), move to a parallel trajectory (cycle), or gradually re-converge to its original path towards a stable fixed point. In contrast, our G_{mn} is obtained by setting a *permanent* perturbation, which does not relax in time, on a selected source node n . Under these conditions, even if the system is at a stable fixed-point $\mathbf{x} = (x_1, \dots, x_N)^T$, the permanent perturbation will force it into a new perturbed state $\mathbf{x} + d\mathbf{x}$, in which *all* nodes have drifted away to some extent from their original state.

In mathematical terms, the linear stability framework introduces a perturbed *initial condition* ($x_n(t = 0) = x_n + dx_n$), while our perturbations constitute a perturbed *boundary condition* ($x_n(t) = x_n + dx_n$, for all t). Hence, these two concepts are indeed related, as correctly identified by the Referee, yet still distinct.

There are two main reasons leading us to focus on permanent perturbations:

Empirical relevance. Most relevant systems reside in the vicinity of a permanent fixed-point, and are hence, in the long term, insensitive to instantaneous perturbations, of the Lyapunov variety. Such perturbations decay in time and very rarely penetrate deep into the network. Under these conditions, the amount of information traversing through a selected node, depends more on the system's stability than on that node's dynamic characteristics. For instance, if the perturbation signal decays rapidly, a distant node would barely even be exposed to the propagating information. *Permanent perturbations, on the other hand, behave as a constant source of propagating information, allowing us to test the role of all network paths in propagating the information flow.*

Practical observations. Permanent perturbations represent a common procedure for controlled observation of many complex systems. For instance, in sub-cellular biology, genetic knockouts – a permanent perturbation - allow us to observe information flow in gene regulation. Similarly, in social systems stubborn agents disseminate information by sticking to their unchanged opinions. Permanent perturbations are also relevant in naturally occurring settings, such as component failure in technological networks, or species extinction in ecological networks – all time invariant perturbations that *force* the system to respond. Hence, from an empirical observation perspective, permanent perturbations represent a highly relevant premise for information flow analysis.

Revisions:

Following this insightful observation by the referee, we added a discussion at the end of **Supplementary Section 1 (Sec. 1.5)** on permanent vs. transient perturbations, along the lines of the above response.

REVIEWERS' COMMENTS:

Reviewer #1 (Remarks to the Author):

The authors have apparently put a lot of efforts revising the paper and arguing its importance, especially through the rebuttal. However, based on the scientific content, I still find the contribution of the paper marginally novel, and does not meet the criteria of publication in Nature Communications.

In the rebuttal letter, the authors argue that the main novelties of the paper are the introduction of “the measure of (information) flow”, and “analytical prediction that exposes how network structure, coupled with nonlinear dynamics, translates into information flow pathways”. First, one can always define a measure for a network, the question is how is such measure relevant. The paper fails to address this important question (simply compute the measure for a collection of systems does not justify the actual relevance of such measure). Simply calling the measure “information flow” also does not solve this problem. Secondly, comparing to a previous publication [17-18], the only conceptual difference is that the current manuscript considers local perturbations when certain nodes are being held constant. Of course, a major issue is that, when being applied to nonlinear systems, there has to be a stable steady state to begin with. The manuscript carefully selected examples of systems that do admit steady states, but the main claim that the results “uncover universal rules that link structure and dynamic information flow”, is way beyond the actual findings.

Reviewer #2 (Remarks to the Author):

I have read the revised version of the paper made by the authors.

I have to say that in the present form I now understood better the point that the authors stressed in their paper. I appreciated very much the extra work they decided to go in order to answer my questions.

One of the main troubles I had was related to the mean field hypothesis. I now acknowledge that the authors make a step further by using the Degree-based Mean field.

Supplementary Section 4 is particularly important, and I believe the paper has been greatly improved by this addition.

The topic treated are of great interest because they extend in a novel way the study of information flow on a complex network. Therefore I strongly suggest publication on your journal.

Reviewer #3 (Remarks to the Author):

I believe the authors have addressed the concerns of the reviewers. Specifically, my concerns have been addressed. I continue to recommend publication of this manuscript.

Reviewer #1

Comment:

The authors have apparently put a lot of efforts revising the paper and arguing its importance, especially through the rebuttal. However, based on the scientific content, I still find the contribution of the paper marginally novel, and does not meet the criteria of publication in Nature Communications. In the rebuttal letter, the authors argue that the main novelties of the paper are the introduction of “the measure of (information) flow”, and “analytical prediction that exposes how network structure, coupled with nonlinear dynamics, translates into information flow pathways”. First, one can always define a measure for a network, the question is how is such measure relevant. The paper fails to address this important question (simply compute the measure for a collection of systems does not justify the actual relevance of such measure). Simply calling the measure “information flow” also does not solve this problem.

Response:

While the *validity* of a scientific contribution is an objective matter, its *significance* is often more subjective, influenced by each individual’s *scientific taste*, which here seems to be the root of the Referee’s reservation. Despite failing to convince Referee 1, we were delighted to see that the other two Referees, already in their first reports, acknowledged our contribution’s importance, stating (Referee 3) that our results are

...of a broad interest and therefore appropriate for Nature Communication’s diverse readership... the results presented here can have a major impact on future works.

Hence, in this subjective test of *significance*, we find that the Referee’s critique is not shared by his/her peers. Moreover, despite its subjective factor, we find the Referee’s dismissal of our suggested measure of Flow, exaggerated and unsubstantiated: we do not *simply call the measure information flow*, as implied in the comment – we formalize what is, in our view, a most natural quantification of flow. To understand this, let us reiterate the rationale of our formalism:

- *Information flow is captured by the propagation of perturbative signals.* This premise is not new. Rather it is fundamental to Statistical Physics, from phase transitions to renormalization group theory, and especially relevant in Network Science, where networks aim to visualize how influence travels from source to target, namely how a source’s perturbation impacts a target’s activity. It also represents the most common empirical access to information flow, *e.g.*, measuring a cell’s response to genetic perturbations.
- *Freezing a network component helps capture its contribution to the flow.* Indeed, it naturally follows from the above that blocking the propagation of a signal through a node/link/pathway can help quantify its specific contribution to the spread of information in the system – precisely the meaning of *freezing*, as defined in our paper.

Therefore we are not just defining an obscure measure, but rather formalizing a most natural and pertinent quantity in the context of network dynamics.

The above explains how our measure of Flow is *insightful and meaningful*, however, in our work we also establish its *relevance*. Indeed, the classes we predict help identify the main arteries through which perturbations, or information signals, spread through the system, thus highlighting the network components that dominate information flow. *Following almost two-decades of extensive work on network centrality measures, obtaining the components that are most dynamically central is no small feat.* Especially, when showing that *topological* centrality (*e.g.*, hubs) does not always translate to dynamic centrality (*e.g.*, degree averting flow). Such analytically derived insight, that information is not always tunneled by the hubs, is a *game-changer* in terms of the common interpretation of structure vs. dynamics.

Finally, we refer the Reviewer to our analysis of epidemic spread, where our measure of Flow provided direct implications on time-evolving immunization strategies, a lacking concept that clearly exemplifies the *relevance* of our framework.

Revisions. We added a few sentences in the Introduction and Discussion sections to further emphasize the significance and relevance of our results.

Comment:

Secondly, comparing to a previous publication [17-18], the only conceptual difference is that the current manuscript considers local perturbations when certain nodes are being held constant.

Response:

What our current paper has in common with the Refs. [17-18] is the notion of observing network dynamics through its response to local perturbations. As explained above this is practically the premise of all Statistical Physics analyses, and hence not *the* conceptual novelty of our current contribution. The concept of Flow, which we introduce here (indeed, by considering the effect of holding certain nodes constant) *is* novel, in that, as opposed to many works which seek the most *influential* nodes, here we focus on the ones that are most efficient *transmitters of influence*. Hence, instead of measuring the impact of a network component as a *source* of information, as, was done, *e.g.*, in [17-18], here we predict the contribution of that network component to the *mediation* between all sources and all potential targets. *This is especially important, as the majority of the time a node is not a source of information, but rather a mediator of information that originates at random network locations, an insight that, to the best of our knowledge is introduced here for the first time through our concept of Flow.*

Of course, our main novelty, as emphasized in the previous report, is not just in introducing this pertinent concept, but primarily in analytically predicting its behaviour from the interplay between the topology and the system's interaction mechanisms, a challenge that, being at the intersection of nonlinear dynamics and complex networks, seldom succumbs to analytical treatment.

Revisions. We now added a paragraph at the end of Sec. II.A where we discuss the conceptual novelty and motivation in focusing on *mediation of information* rather than on direct influence.

Comment:

Of course, a major issue is that, when being applied to nonlinear systems, there has to be a stable steady state to begin with. The manuscript carefully selected examples of systems that do admit steady states, but the main claim that the results “uncover universal rules that link structure and dynamic information flow”, is way beyond the actual findings.

Response:

We strongly disagree with this Comment:

- We did not *carefully select examples*, but rather used well-established models from a spectrum of scientific domains. Our only *selection criterion* was to have a balanced representation of the three classes we predict, namely to have models with ω positive, negative and zero. Without this *presentational* constraint, Eq. (1) allows for a broad variety of dynamic models – a level of generality, that even if not all encompassing – is still rarely encountered in the context of network dynamics.
- The Referee seems to ignore our analysis of the SIR epidemic model, which captures the evolution of flow patterns as the system transitions from the *unstable healthy state* to its final pandemic state, representing a case of Flow, which is out of equilibrium, originating in an unstable fixed-point. This example was indeed *carefully selected*: for its relevance, for its exposure of the novelty of temporally evolving flow patterns, and for the fact that it captures Flow around an *unstable* steady state.

Reviewer #2

I have read the revised version of the paper made by the authors.

I have to say that in the present form I now understood better the point that the authors stressed in their paper. I appreciated very much the extra work they decided to go in order to answer my questions.

One of the main troubles I had was related to the mean field hypothesis. I now acknowledge that the authors make a step further by using the Degree-based Mean field.

Supplementary Section 4 is particularly important, and I believe the paper has been greatly improved by this addition.

The topic treated are of great interest because they extend in a novel way the study of information flow on a complex network. Therefore I strongly suggest publication on your journal.

Response:

We wish to thank Referee 2 for these positive comments and for his/her thorough report that truly helped us improve the paper, its presentation, validation and scope.

Reviewer #3

I believe the authors have addressed the concerns of the reviewers. Specifically, my concerns have been addressed. I continue to recommend publication of this manuscript.

Response:

We wish to thank Referee 3 for his/her constructive report and appreciative assessment.